# Versatile whole-organ/body staining and imaging based on electrolyte-gel properties of biological tissues

Etsuo A. Susaki ⬡ et al.#

Whole-organ/body three-dimensional (3D) staining and imaging have been enduring challenges in histology. By dissecting the complex physicochemical environment of the staining system, we developed a highly optimized 3D staining imaging pipeline based on CUBIC. Based on our precise characterization of biological tissues as an electrolyte gel, we experimentally evaluated broad 3D staining conditions by using an artificial tissue-mimicking material. The combination of optimized conditions allows a bottom-up design of a superior 3D staining protocol that can uniformly label whole adult mouse brains, an adult marmoset brain hemisphere, an ~1 cm$^3$ tissue block of a postmortem adult human cerebellum, and an entire infant marmoset body with dozens of antibodies and cell-impermeant nuclear stains. The whole-organ 3D images collected by light-sheet microscopy are used for computational analyses and whole-organ comparison analysis between species. This pipeline, named CUBIC-HistoVIsion, thus offers advanced opportunities for organ- and organism-scale histological analysis of multicellular systems.

#A full list of authors and their affiliations appears at the end of the paper.

Since German anatomist Walter Spalteholz developed the first tissue clearing reagent over 100 years ago, systematic three-dimensional (3D) observation and analysis of whole organs and whole bodies have been continuously conducted in biomedical research, linking classical anatomy to modern systems biology[1,2]. With the advent of state-of-the-art tissue clearing and 3D imaging methods, a large volume of samples can be viewed comprehensively with cellular to subcellular resolution over single organs and organisms (recently reviewed in Ueda et al.[3]). Collecting biological information requires appropriate labeling according to structure, cell type, and cell activity. Various genetic and viral tools applied for this purpose have facilitated numerous discoveries[2,4]. In addition, whole-mount staining with clearing has been assessed, starting in the 1980s with the insect and shrimp nervous systems and the Xenopus embryo[5–7]. Recently, whole-mount staining with clearing has been expanded to the 3D observation of murine and human embryos[8,9], various animal organs and bodies[10–19], and human pathological specimens[12,20–24]. Staining is more advantageous than genetic or viral tools in terms of (1) applicability to a broad range of samples, including human and non-model animal specimens, and (2) the relative ease of multitarget labeling.

There have been several approaches designed to improve the penetration of stains and antibodies in a large tissue sample. Intensive permeabilization procedures to increase the pore size of fixed tissue have been attempted, including delipidation (sometimes with tissue clearing)[12,13,15,16,21], dehydration[7,8,10,11,17], weaker fixation[10], and partial digestion with proteases[10,11]. Urea or SDS was introduced to control the binding affinity of stains and antibodies during penetration[15,25]. Several physical methods, such as electrophoresis and pressure, were tested on acrylamide-embedded samples[26,27].

However, the insufficient penetration of stains and antibodies remains a crucial bottleneck in many 3D staining cases. Researchers often face a situation where even small dyes do not penetrate 3D samples, implicating the complex physicochemical environment in the staining system. Currently, 3D staining mainly uses a small variety of stains and antibodies on samples of relatively small and thin tissues, partially dissected tissues, or embryonic tissues with little extracellular matrix. Low-density antigens such as c-Fos, amyloid plaques or microglia markers have demonstrated the capacity for homogeneous staining of sizeable tissue (e.g., on the order of $cm^3$ for whole adult mouse brain or dissected human specimens)[17,28]. On the other hand, higher density antigens, such as NeuN and neurofilament, have not yet been adequately demonstrated to be capable of such staining. Other attempts have also been made, including the iterative supply of staining reagents[12,21] or the use of a specialized device[26] or transcardial perfusion-based staining[16,18].

To overcome these limitations, we applied an objective rather than an empirical approach to exploring essential 3D staining conditions. Beginning with a detailed characterization of biological tissue by material chemistry methods, we found fixed and delipidated tissue for optical clearing to be equivalent to an electrolyte gel of cross-linked polypeptides. Then, inspired by a diffusion-reaction scheme and the characterization of biological tissue as a gel, we broadly evaluated 3D staining conditions for the bottom-up design of a superior protocol. By using CUBIC-HistoVIsion for 3D staining, tissue clearing, and volumetric imaging, we successfully stained and imaged whole adult mouse brains, a whole adult marmoset brain hemisphere, and an ~1 cm³ tissue block of a postmortem adult human cerebellum with various cell-impermeant nuclear stains and antibodies. We also showed that the whole-organ structural information given by nuclear staining enables the registration and alignment of multiple whole-organ images. Furthermore, the signal-to-background ratio (SBR) of these 3D images was high enough for the computational detection of labeled cells. These results highlight the perspective of systematic quantification and analysis of biological information from whole-organ and whole-body 3D staining and the subsequent computational processes.

## Results

**Fixed and delipidated tissue is an electrolyte gel.** To develop a bottom-up design of an optimized 3D staining protocol, we attempted to determine the physicochemical properties of paraformaldehyde (PFA)-fixed and delipidated tissue, which had undergone tissue clearing with the CUBIC protocol (see "Methods" for details). Here, we used the brain as a representative organ. We found that the sample could repeatedly and reversibly swell and shrink under various chemical conditions (Fig. 1a). This phenomenon provided substantial evidence that the tissue sample could be characterized as a gel: in materials science, gels (1) have mesh/network structures (introduced by PFA fixation), (2) do not dissolve in a medium (which is self-evident), and (3) have the potential for repeated and reversible swelling and shrinkage[29]. This notion is consistent with the early discovery by Tanaka, who documented that biomaterials such as cross-linked proteins, agarose and DNA can be considered gels[30].

We then narrowed down the specific physicochemical properties of the sample in terms of its characterization as a gel. We first investigated changes in tissue composition during the clearing procedure. The process was highly coordinated with the reduction in phospholipids and cholesterol and complementary increase in water content, while the protein amount was mostly conserved (Fig. 1b; Supplementary Fig. 1a, b). Using these data and extrapolating from previous reports on the whole rat brain[31], we estimated changes in tissue composition before and after delipidation (Fig. 1c). These results suggest that the brain is a gel mainly composed of cross-linked proteins.

We then applied small-angle X-ray scattering (SAXS) analysis to the delipidated brain. This analysis aimed to measure the nanometer-scale structure of the sample (on the order of $1-10^2$ nm) in the form of the scattering intensity $I$ (the Fourier transform of the paired electron density distribution inside the sample) as a function of the magnitude of the scattering vector $\mathbf{q}$ (which is inversely proportional to the size of a scattering structure[32]) (Fig. 1d).

The following findings from the acquired $I$-$q$ profiles revealed the structural and chemical characteristics of the delipidated brain. First, the scattering peak at $q \approx 0.08$ Å$^{-1}$ in the nondelipidated brain disappeared after delipidation (Supplementary Fig. 1c–e). The corresponding d-spacing based on Bragg's Law ($d = 2\pi/q$) is ~8 nm. As suggested in a previous SAXS study of the human brain[33], this peak is likely attributed to the phospholipid layers of the myelin sheath (whose thickness is on the order of ~1–10 nm[34]). Hence, the disappearance of the scattering peak again confirmed that the delipidation procedure had indeed removed the lipid contents from the sample (Supplementary Fig. 1f). Second, the $I$-$q$ profile showed a broad peak ($q \approx 0.02-0.04$ Å$^{-1}$, ~15–30 nm) when the buffer had lower ionic strength, while the peak disappeared at high NaCl concentrations (arrowheads in Fig. 1e, f). The peak thus might reflect long-range spatial correlations among polymers at this order of distance that depend on ionic interactions. Such a salt-dependent peak is commonly observed in electrolyte gels and implies changes in ionic interactions between polymer molecules that are affected by the ionic strength of the solution[35]. Therefore, the delipidated brain can be characterized as an electrolyte gel (Supplementary Fig. 1f). Since most of the cationic amino residues are masked by PFA fixation and the isoelectric point of the CUBIC-delipidated brain is approximately pH 5[36], the dominant ionized residues in the polymer are thought to contain an anionic carboxyl group.

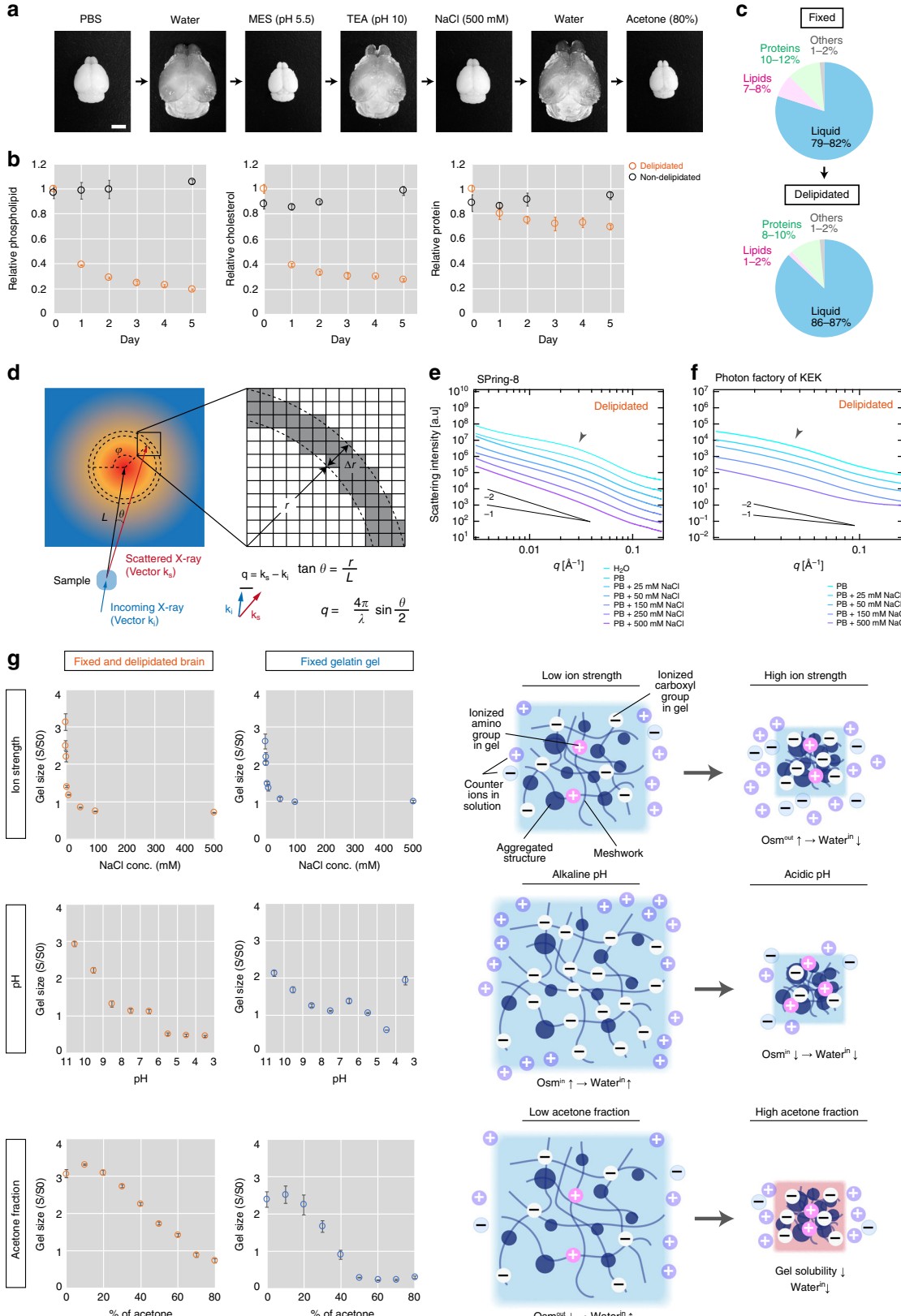

The SAXS analysis further implied that the brain sample could be characterized as a gel with a fractal nature. In general, the $I$-$q$ profiles of highly heterogeneous gels exhibit the following power-law behavior:

$$I \propto q^{-D},\tag{1}$$

where $D$ is the dimension of the mass fractal[37]. The acquired $I$-$q$ profiles (Fig. 1e, f) indeed demonstrated power-law behavior with a slope of $D \approx 2$, reflecting a highly heterogeneous gel structure with a fractal nature[32,37].

The swelling-shrinkage curves of gels[30,35] further indicated that the delipidated brain responds to the surrounding ion strength,

**Fig. 1 Fixed and delipidated tissue is an electrolyte gel. a** A fixed and delipidated whole mouse brain treated with different chemical conditions. MES: 2-(N-morpholino)ethanesulfonic acid, TEA: triethanolamine. Scale: 5 mm. **b** Changes in the components inside the tissue during the delipidation procedure. The values were calculated as a ratio to the average of the delipidated samples (day 0) and indicate the means ± SD of the ratios ($n = 3$ biologically independent samples). **c** Changes in tissue components before and after delipidation. **d** Schematic of SAXS measurement and analysis. The relationship between the intensity of the scattered X-rays ($I$) and the scattering angle ($\theta$) reflects nanometer-order structures inside the sample. The value of $I$ in terms of the width $\Delta r$ was calculated along the radius line ($r$) on the 2D intensity plot at a specific angle ($\varphi$) [or the circular average ($\varphi = 0–2\pi$)]. The scattering angle ($\theta$) is transformed to the magnitude of the scattering vector **q** (which is inversely proportional to the size of the inside structure) according to the equation in the panel. $L$: distance between the sample and the detector, $\lambda$: wavelength of the X-ray. **e, f** $I$–$q$ plot of the SAXS analysis of the brain slices for different NaCl concentrations, measured at SPring-8 (**e**) and the Photon Factory of High Energy Accelerator Research Organization (KEK) (**f**). The values on the y-axis were shifted by multiplying by shift factors (in **e**, 1000, 500, 80, 30, 10, 2, and 1 for $H_2O$, PB, PB with 25 mM, 50 mM, 150 mM, 250 mM, and 500 mM NaCl, respectively; in **f**, 35, 7, 4, 6, and 1 for PB, PB with 25 mM, 50 mM, 150 mM, and 500 mM NaCl, respectively). Lines indicate the slopes of the mass fractal dimension $D$. **g** Swelling-shrinkage curves of the delipidated tissue and fixed gelatin gel. The y-axis ($S/S0$) represents the relative value of the gel area (mean ± SD, $n = 3$, biologically independent samples).

pH and hydrophobic solvent fraction (Fig. 1g). An increased ionic strength in the buffer induced sharp shrinkage of the tissue, suggesting that the delipidated brain could characterized as a typical ionized electrolyte gel[35]. The multistate swelling and shrinkage of the brain under changing pH also suggests that the behavior is primarily affected by the degree of ionization of the carboxyl group of the polypeptide polymer. We also observed more than fourfold shrinkage in the area (corresponding to an eight-fold decrease in volume) resulting from the decrease in gel solubility as the acetone fraction increased.

We compared the swelling-shrinkage behavior of the brain with that of several artificial gels, including PFA-fixed type-B gelatin gel, PFA-fixed agarose gel, and polyacrylamide gel as representatives of ionized polypeptide gels, nonionized natural polymer gels, and artificial hydrophilic gels, respectively (Fig. 1g; Supplementary Fig. 1g). Under our experimental conditions, the nonionized agarose and polyacrylamide gels did not respond to ionic strength and pH changes. Only the polyacrylamide gel responded to the acetone fraction with a limited dynamic range. On the other hand, the swelling-shrinkage behaviors of gelatin gel under various ionic strengths, pH values, and acetone fractions were similar to those of the delipidated tissue. We note that the gelatin gel exhibited a more dynamic volume change than the tissue under pH change. Similar dynamics were observed in the case of an artificial gel with amphoteric ions[38]. Moreover, the gelatin gel was more sensitive to the increase in the acetone fraction. Various factors, such as the complexity of the gel components, the rigidity of the gel, the ionic charge state, the degree of ionic imbalance (Donnan's membrane equilibrium), or internal interactions, may account for the differences in the macroscopic size change patterns of these gels. Nonetheless, the overall swelling-shrinkage behaviors of gelatin gel, a typical example of polypeptide gels, were closest to those of the tissue, with slight quantitative differences.

Given the commonality of tissue composition (lipids, proteins, nucleic acids, and water; Fig. 1c), we assumed that other delipidated tissues could also be modeled as an electrolyte gel with similar properties to those of the brain. Indeed, repeated and reversible swelling-shrinkage behaviors of delipidated kidney, liver, and muscle tissue (Supplementary Fig. 1h), similar to those seen in the brain (Fig. 1a), were observed. In particular, (1) expansion in the alkaline pH range, (2) shrinkage in a highly ionized state, and (3) shrinkage again under a high acetone fraction proved the consistency of our electrolyte gel model of biological tissue.

Taken together, we conclude that tissue that was fixed and delipidated for optical clearing could be generally characterized as an electrolyte gel composed primarily of cross-linked polypeptides, and the chemical properties are very similar to those of an artificial protein gel. This conclusion is also consistent with our previous findings[36].

**Developing versatile whole-organ/body dye staining**. Based on these unveiled electrolyte gel properties of delipidated tissue, we considered a simple theoretical model as a first-order approximation of the 3D staining process, in which Fick's law governs the diffusion of reactive solute into an electrolyte gel[39]. The reaction-diffusion process is written as

$$\frac{\partial C}{\partial t} = D_{\text{diff}} \cdot \frac{\partial^2 C}{\partial x^2} + R, \tag{2}$$

where $C$, $t$, $D_{\text{diff}}$, and $x$ indicate solute concentration, time, the diffusion constant, and position, respectively. $R$ denotes the reaction term: to describe the interaction of the solute and its interacting target(s) in the gel, we chose the reversible one-to-one binding and unbinding scheme with parameters $k_{\text{on}}$ and $k_{\text{off}}$ (see "Methods" for details). The value of $D_{\text{diff}}$ for the gel can be affected by various factors, such as temperature ($T$), the gel mesh structure ($\eta$) and particle size ($r$), as described by the formula for simple diffusion (Einstein-Stokes law):

$$D_{\text{diff}} = f(T, \eta, r). \tag{3}$$

To predict the possible contribution of these parameters in the gel staining, we first simulated the staining patterns of a circular 2D gel (Supplementary Fig. 2a, b). The simulation predicted that lower $D_{\text{diff}}$, higher $k_{\text{on}}$, a lower solute concentration, or a higher concentration of interaction target in a gel would retain the staining signal at the edge of the gel (Supplementary Fig. 2a). For convenience, we categorized the simulation results as (1) rimmed, suggesting the solute is trapped on the gel surface and the signal remains at the edge of gel, or (2) gradual, indicating improved penetration efficiency. Note that these patterns are the two extremes and that continuous changes in these parameters would produce intermediate patterns.

We then broadly examined the experimental conditions that could affect these parameters. To perform the test in a semiquantitative manner and in a controlled setting, we used fixed gelatin gel as a tissue surrogate. We stained cylinder-shaped, PFA-fixed gelatin gels containing DNA or rabbit immunoglobulin with a nuclear stain or an Alexa Fluor dye-conjugated anti-rabbit secondary antibody, respectively. Next, we determined the staining pattern in the middle of the gel according to the simulation-based criteria (Fig. 2a).

We first tested positively charged nuclear stains. Given the negatively charged environment of fixed gelatin gels under neutral to alkaline pH conditions, we hypothesized that the strong ionic interactions with the gel would inhibit the penetration of these positively charged dyes, which could be attenuated by increasing the salt concentration.

As expected, the staining pattern for the ionized dye propidium iodide (PI) changed drastically from the rimmed to the gradual pattern by increasing the salt concentration (e.g., 500 mM NaCl

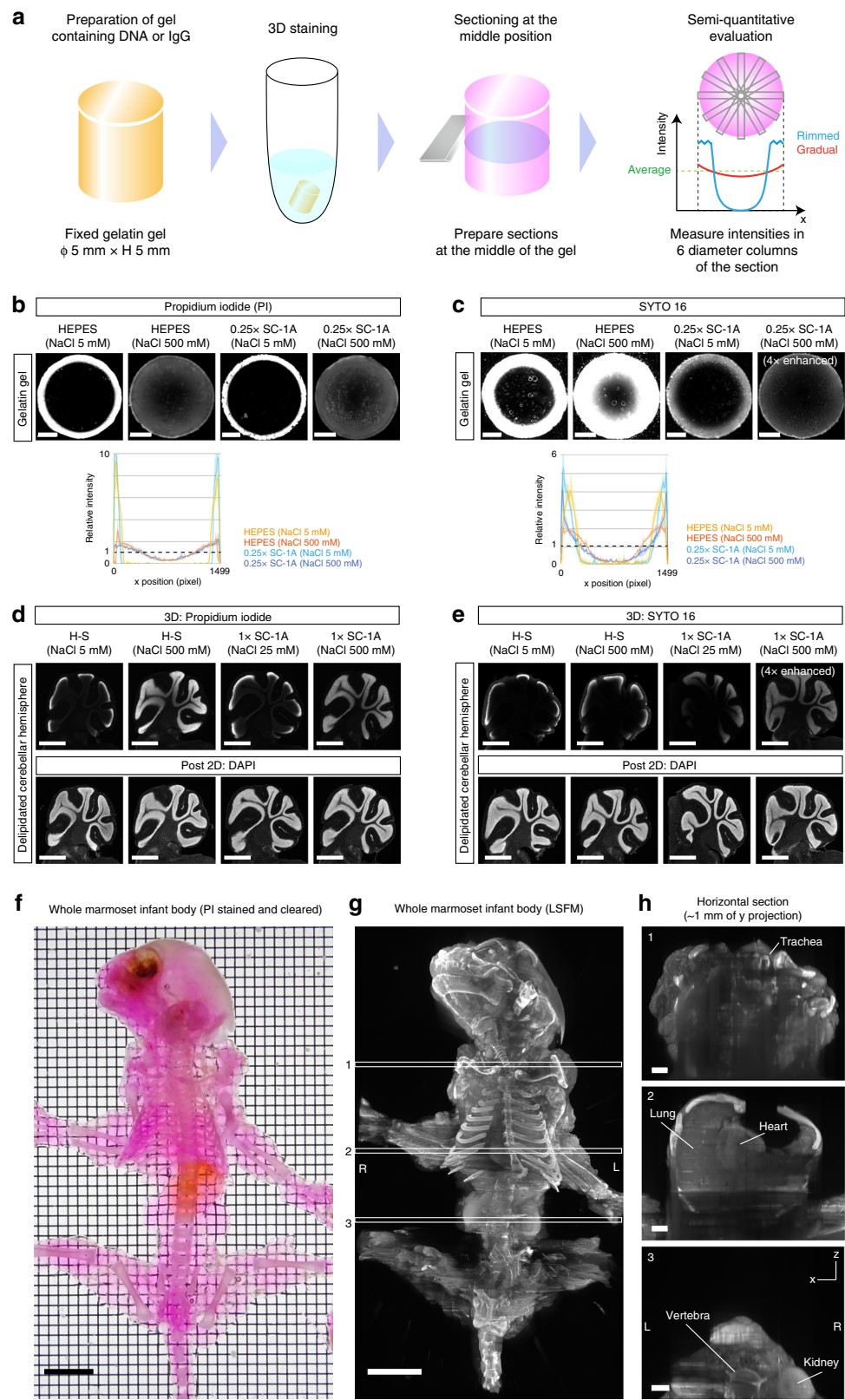

in HEPES buffer) (Fig. 2b). This range of NaCl concentrations potentially cancelled the ionic interaction of the fixed gelatin gel (Fig. 1e–g). Shrinkage of the gel under high salt conditions (Fig. 1g) did not affect penetration, suggesting that the size of the dye molecule was sufficiently small regardless of the volume phase of the gel. On the other hand, when tested with another

ionized but more lipophilic dye, SYTO 16, the rimmed pattern was still observed for HEPES buffer containing 500 mM NaCl (Fig. 2c). Therefore, given our previous success in 3D staining using SYTO 16 with Sca*le*CUBIC-1[13], we tested a Sca*le*CUBIC-1-based staining buffer. When supplied with 500 mM NaCl, the reagent indeed changed the SYTO 16 staining result from the

**Fig. 2 Development of a versatile protocol for whole-organ/body dye staining. a** Schematic of the surrogate assay. A cylinder-shaped, fixed gelatin gel containing DNA or rabbit immunoglobulin was used to experimentally evaluate the 3D staining patterns in the interior of the gel (rimmed or gradual patterns). **b** Representative results of DNA-containing gelatin gel staining with propidium iodide (PI), an ionized, lipophobic, and cell-impermeable nuclear stain. A high concentration of salt modulated the gel–stain interaction, leading to the improvement of the interior staining patterns. Scale: 1 mm. **c** Representative results of DNA-containing gelatin gel staining with SYTO 16, a lipophilic and cell-permeable nuclear stain. Additional use of ScaleCUBIC-1A (SC-1A) chemicals in high-concentration salt conditions improved the penetration patterns with decreased signal intensities (indicated as "4× enhanced"). Scale: 1 mm. The profiles in **b** and **c** show the mean intensity ± SD of the six diameter regions as in **a**. **d**, **e** The results in **b** and **c** were replicated by using the fixed and delipidated tissues (cerebellar hemispheres). After 3D staining, the samples were sectioned to evaluate the infiltration of the stain inside the tissue. The sections were also restained with DAPI to check the detectability of the nuclei (post 2D). Scale: 1 mm. **f–h** The interaction-modulated 3D staining was scaled up to a whole infant marmoset body. **f** The cleared and PI-stained infant marmoset body. PI staining was performed under a high ion strength condition (see "Methods"). Scale: 1 cm. **g** The reconstituted whole-body 3D image acquired with a custom-built LSFM (voxel size: 10.3 × 10.3 × 100 μm$^3$) after staining and clearing. White boxes indicate the positions of the reconstituted x–z images in the panels on the right. Scale: 1 cm. **h** The reconstituted x–z images at the indicated positions in **g**. L left, R right. Scale: 2 mm.

rimmed to the gradual pattern (Fig. 2c). This result implies that the addition of ScaleCUBIC-1 reagent can further modulate the interaction of SYTO 16 with the gel.

We also tested the case of a negatively charged dye, eosin. The 3D staining time was much shorter (~4 h for the cerebral hemisphere) than in the case of nuclear stains (2–3 days for a similar-sized sample). The dye rapidly penetrated the gelatin gel independently of the salt concentration, suggesting no absorption at the gel surface due to ionic interactions. The ScaleCUBIC-1 reagent reduced the penetration efficiency, unlike the case with the positively charged dye (Supplementary Fig. 2c).

We then applied these findings to stain delipidated tissues. Homogeneous nuclear staining on these samples was observed for the optimal staining conditions found in the gelatin gel (Fig. 2d, e), verifying our assumption that the fixed gelatin gel could be considered a tissue surrogate. To further test the scalability of the single-step 3D staining achieved by modulating gel–solute interactions with additives, we stained a whole infant marmoset body with PI in a high-concentration salt buffer (0.1 M PB with 1.5 M NaCl) (Fig. 2f). After tissue clearing and whole-body staining, the sample was imaged with a custom-built macrozoom light-sheet fluorescence microscope (LSFM) covering the whole animal (Fig. 2g). The acquired images indicated the successful staining of the entire body with a 3D-staining condition that modulates gel–solute interactions (Fig. 2g, h; Supplementary Movie 1).

ScaleCUBIC-1 with 500 mM NaCl buffer reproducibly supported efficient 3D staining of various nuclear stains in addition to PI and SYTO 16 (Supplementary Fig. 2d). Therefore, we finally employed the recipe as a general 3D nuclear staining buffer in the following experiments. However, as indicated by the results of the gelatin gel assay, more rapid 3D staining was achieved without additives for the negatively charged dye eosin (Supplementary Fig. 2e).

**Developing versatile whole-organ immunostaining.** We next sought to determine the experimental conditions for whole-organ antibody staining with a surrogate gel assay. Since the molecular weight of the antibody is more than 10$^3$-times larger than that of small dyes and the antibody would potentially interact with the electrolyte gel in a more complicated manner, we determined that a broader range of experimental conditions should be examined than for the small molecule dyes. First, we tested whether the gel–solute interactions validated by the nuclear stains (Fig. 2) were also essential for 3D immunostaining. We found that the higher the antigen (rabbit immunoglobulin) concentration in the gel, the greater the interaction between an antibody and the antigen-containing gel increased, resulting in the rimmed staining pattern (obvious at 0.5 μg/mL, Fig. 3a).

The results prompted us to expand the exploration of essential conditions that could influence the 3D immunostaining results. As suggested previously[10], the gelatin gel concentration had an

impact on the staining pattern (the rimmed pattern was obvious at 10–15%, Fig. 3b), possibly by altering the gel mesh structure and the nonspecific interactions between the antibody and the gel. The Alexa Fluor dye types somewhat affected the penetration efficiency, probably due to differences in the gel–Alexa Fluor interactions (Supplementary Fig. 3a). The temperature during staining and the initial concentration of antibody in the buffer determined the efficiency of penetration (Supplementary Fig. 3a), as suggested by the diffusion-reaction simulation. The rimmed pattern was observed at 4 °C or at an antibody concentration of 2.5 μg/mL, indicating the appropriate temperature range (>room temperature) and antibody concentration (~>5 μg/mL) for 3D immunostaining. On the other hand, the NaCl and detergent concentrations did not affect the penetration efficiency but rather changed the signal intensity (Supplementary Fig. 3a). We note that, as in the case of nuclear stains (Fig. 2), the ionic strength required to saturate the shrinkage of the gel (Fig. 1g) did not prevent antibody penetration. We also note that the increased immunostaining signal under high concentrations of Triton X-100 was not attributable to increased antibody penetration into the gel but to increased antibody binding. Indeed, 10% Triton X-100 reproducibly increased the signal intensities from tissue section staining compared with 0.1% Triton X-100. (Supplementary Fig. 3b). Hence, we applied 200 mM NaCl and 10% Triton X-100 to our final immunostaining buffer.

We validated some of these essential conditions using delipidated brain samples. The concentrations of Quadrol and urea used in nuclear staining (Fig. 2) were again effective in tissue 3D immunostaining, presumably as a result of attenuating antibody–tissue interactions (Supplementary Fig. 3c). Furthermore, the results in Fig. 3a suggest that conventional two-step staining with primary and secondary antibodies would not be useful in 3D staining unless the target antigens were distributed sparsely. This is because, when the target is abundant, a large amount of the primary antibody fills the gel as a dense antigen of the secondary antibody, thus reducing its penetration efficiency. This assumption was confirmed by 3D immunostaining against NeuN. The one-step procedure using either the dye-conjugated primary antibody or the complex of the primary antibody and secondary Fab fragment successfully supported uniform 3D staining in the tissue sample (Fig. 3c, upper panels). On the other hand, a two-step procedure using primary and secondary antibodies resulted in limited signal in the sample (Fig. 3d, upper panels). Since post-cut 2D staining confirmed that the NeuN antigen was well preserved and that the primary mouse anti-NeuN antibody had efficiently penetrated the tissue (Fig. 3c, d, lower panels), this difference was due to insufficient penetration of the secondary antibody.

Inspired by the results in Fig. 3b and those of a previous study[10], we also tested partial digestion of the extracellular matrix

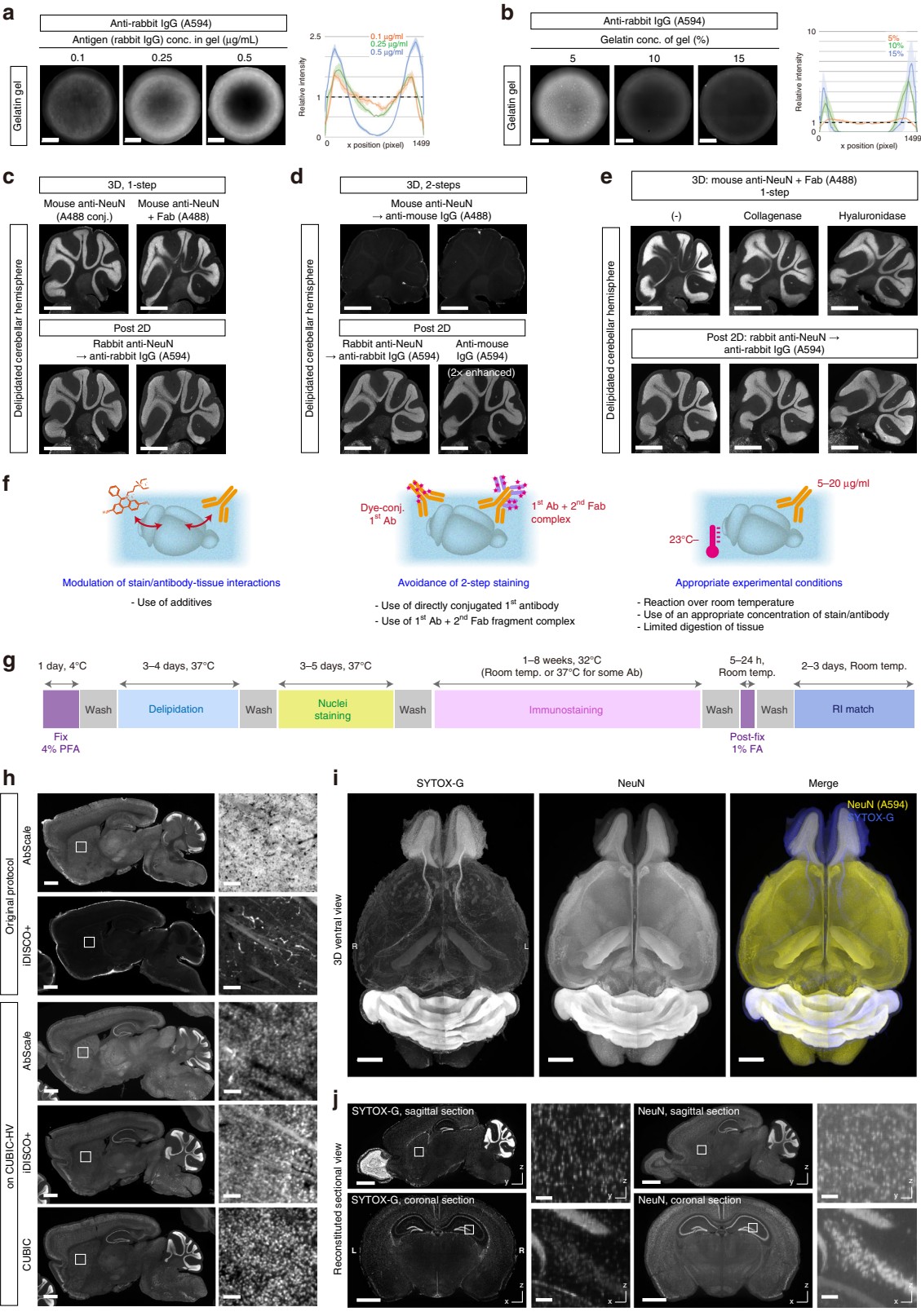

(ECM). We treated the tissue with collagenase or hyaluronidase under highly stringent conditions to achieve limited digestion. The treatment indeed ameliorated the edge binding of the antibody (Fig. 3e).

Taken together, we postulated the essential conditions for an efficient 3D staining protocol as follows: (1) modulation of the interactions between dye/antibody and tissue by chemical additives such as Quadrol and urea, (2) avoidance of two-step staining, and (3) selection of appropriate temperature, ionic strength, detergent concentration, and antibody concentration, and limited digestion of tissue (Fig. 3f). We finally constructed a versatile protocol for 3D histology and volumetric imaging on CUBIC (CUBIC-HistoVIsion version 1.0, hereafter designated CUBIC-HV) by integrating these conditions (Fig. 3g). In the

**Fig. 3 Development of a versatile protocol for whole-organ immunostaining. a, b** Representative results of rabbit IgG-containing gelatin gel staining with anti-rabbit secondary IgG (A594). Scale: 1 mm. The profiles show the mean intensity ± SD of the six diameter regions as in Fig. 2a. **c–e** The results in **a** and **b** were replicated with the fixed and delipidated tissues. **c, d** One-step procedure with mouse anti-NeuN-A488 or unconjugated anti-NeuN with secondary Fab fragment [3D, **c**] and conventional two-step procedure using secondary anti-mouse IgG after the primary antibody [3D, **d**] were compared. Three-dimensional staining results were visualized with A488. After 3D staining, the sample was sectioned and restained with rabbit anti-NeuN antibody followed by secondary anti-rabbit IgG (A594) or restained only with secondary anti-mouse IgG (A594) to confirm the detectability of NeuN antigen or the infiltration degree of primary anti-NeuN antibody, respectively (post-2D). Scale: 1 mm. **e** Collagenase or hyaluronidase treatment was tested by similar staining procedures. Scale: 1 mm. **f** Summary of essential conditions for efficient 3D staining. **g** Overview of the clearing, staining and RI matching of the CUBIC-HV protocol (ver. 1.0). The indicated durations are assumed for a whole adult mouse brain. Also see Supplementary Fig. 4a. **h** Comparison and modification of previous 3D staining protocols based on CUBIC-HV. Adult mouse brain hemispheres were 3D stained with anti-NeuN antibody (Supplementary Fig. 4d). The intensity range was normalized in each image. Scale: 1 mm (for entire sagittal sections) or 50 μm (for enlarged images). **i** Two-color whole-organ staining and volumetric imaging results of the adult mouse brain with custom-built LSFM. Voxel size: $8.3 \times 8.3 \times 9$ μm³. Three-dimensional images were reconstituted with Imaris software. L left, R right. Scale: 2 mm. **j** Reconstituted sagittal ($y$–$z$) or coronal ($x$–$z$) images of the data in **i**, showing uniform 3D staining and imaging with almost isotropic cellular resolution. The resliced images were prepared with Fiji/ImageJ. Scale: 2 mm (for entire images) or 100 μm (for enlarged images).

CUBIC-HV protocol, we applied Sca*le*CUBIC-1A supplied with 500 mM NaCl for 3D nuclear staining. We also used ionic strength-controlled HEPES buffer (200 mM NaCl) containing 10% Triton X-100 and 0.5% casein for 3D immunostaining. This buffer improved the SBR compared with a conventional PBST buffer with serum (Supplementary Fig. 3d). Limited enzymatic digestion (hyaluronidase or collagenase P) or low levels of additives (Quadrol and urea) can optionally be applied when the dye/antibody penetration is insufficient. We confirmed that the enymatic digestion steps did not quench the GFP signals (Supplementary Fig. 3e). Therefore, we used these enzymes in most of the following experiments for demonstration (Supplementary Fig. 4a). However, the application should be carefully considered.

CUBIC-HV demonstrated 3D staining performance with the cell-impermeable dye SYTOX-G and anti-NeuN antibody comparable to the performance of 2D staining (Supplementary Fig. 4b, c). We further compared previously reported 3D clearing-staining protocols AbSc*ale*[15] and iDISCO+[17] with CUBIC-HV. When anti-NeuN 3D staining of brain hemispheres was performed according to the original protocols (Supplementary Fig. 4d), the expected staining pattern was not completely observed inside the sample (Fig. 3h, Original protocol). Therefore, we modified the protocol by integrating the conditions adopted in CUBIC-HV (Supplementary Fig. 4d). The staining performance was obviously improved and became comparable to that of CUBIC-HV (Fig. 3h, CUBIC-HV). These results suggest the general importance of the 3D staining conditions found in our bottom-up approach. Although we also tested SWITCH-mediated antibody labeling (Fig. 6C in ref. [25]), no expected NeuN signal was observed, at least from our experiment (Supplementary Fig. 4e, f).

Finally, we examined whole-brain 3D staining with the nuclear stain SYTOX-G and anti-NeuN antibody bound with secondary Fab-A594. The stained sample was cleared with refractive index (RI) matching and was used for whole-organ 3D imaging with a custom-built macrozoom LSFM (Fig. 3i, j). We confirmed a uniform staining result on 3D-reconstituted, resliced images with a nearly isometric voxel size ($8.3 \times 8.3 \times 9$ μm³) (Fig. 3j). According to the NeuN data, there was no apparent absorption of light-sheet illumination through the tissue (Supplementary Fig. 4g).

**CUBIC-HV is applicable to various antibodies and dyes**. We applied CUBIC-HV to various markers of particular interest in neuroscience, including neuronal and glial cell markers, axon, dendrite, and synaptic molecules, and vessel markers (Supplementary Data 1). We finally selected 25 out of 44 antibodies listed as suitable for the following whole-organ immunostaining,

according to the macroscopic staining pattern, penetration efficiency, and signal intensity (see "Methods" for details). We also summarized the 2D staining results of the tested antibodies and the corresponding reconstituted sections of the whole-brain 3D staining images (Supplementary Fig. 5a, b).

A series of whole-organ 3D staining and volumetric imaging with the selected antibodies and either green (SYTOX-G) or blue-cyan (BOBO-1) nuclear stains were performed (Fig. 4a; Supplementary Movie 2). A complex of primary antibody and A594-conjugated secondary Fab fragment was used for the one-step staining procedure. The stained and optically cleared adult whole mouse brains were successfully imaged using the custom-built macrozoom LSFM with nearly isotropic resolution ($8.3 \times 8.3 \times 9$ μm³). In the 3D stained and cleared samples, the cellular-to-subcellular structures were reasonably preserved (Supplementary Fig. 6a–n; also see Supplementary Discussion). We also stained a whole mouse kidney with SYTOX-G and an antibody against the pan-expressing transcription factor CREB. The uniform staining result inside the kidney was obtained as in the whole-brain staining (Supplementary Fig. 7a), supporting the applicability of CUBIC-HV to other organs.

We then evaluated whether these whole-brain images with structural information revealed by nuclear labeling could be processed by computational multibrain registration and alignment in the CUBIC pipeline[13,40]. Nine individual whole-brain antibody staining datasets [phosphoneurofilament (phospho-Nf), calbindin D28K, parvalbumin (PV), somatostatin (Sst), choline acetyltransferase (ChAT), dopamine β-hydroxylase (Dbh), tyrosine hydroxylase (Th), tryptophan hydroxylase 2 (Tph2), and copeptin, together with either SYTOX-G or BOBO-1] were successfully registered and aligned against whole-brain anti-NeuN and SYTOX-G staining data (Fig. 4b). The resulting data provided a fused pseudo-multitarget whole-brain antibody staining image (Fig. 4b; Supplementary Movie 3). We further assessed the accuracy of registration among multiple brains by checking the overlap of the corresponding signals. Copeptin and NeuN signals were regionally merged in the hypothalamic paraventricular nucleus (PVN) and the supraoptic nucleus (SON); both Dbh and Th signals were found in the locus coeruleus of the pons; and both PV and calbindin D28K signals overlapped in the cerebellar Purkinje cell layer (Fig. 4b).

Taken together, these results indicate that CUBIC-HV enables stable whole-organ 3D staining, imaging, and computational processing with various dyes and antibodies.

**CUBIC-HV allows multichannel whole-organ labeling**. Compared with genetic tools, histology can more facilitate the staining and imaging of multiple targets in a single specimen (Fig. 5a). The

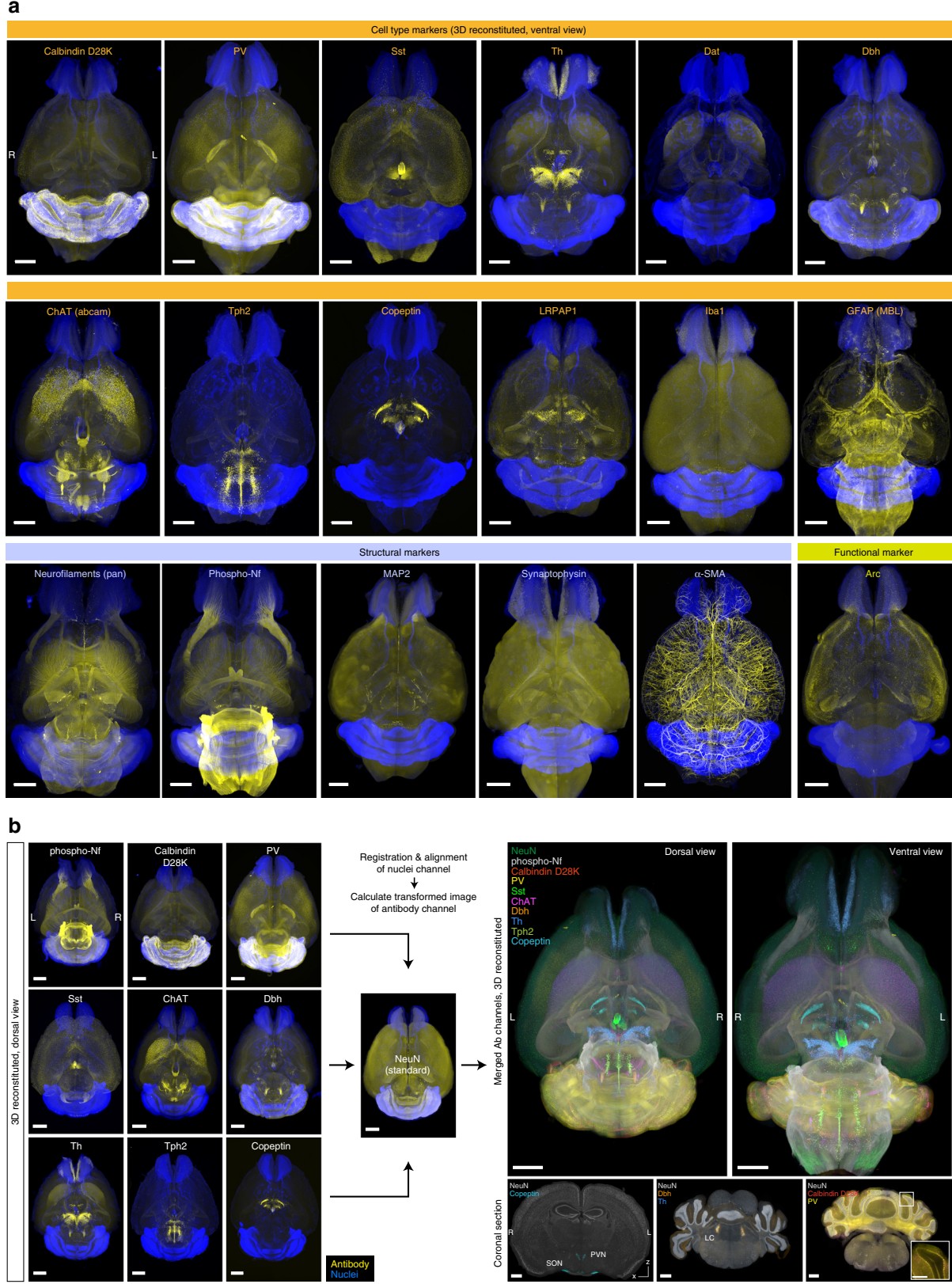

3D histology realized by CUBIC-HV allowed successful multi-color staining of whole brains with a combination of BOBO-1 and three different antibodies related to GABA neurons [mouse monoclonal anti-PV IgG$_1$/Fab-Cy3, rat monoclonal anti-Sst IgG$_{2b}$/Fab-A594 and mouse monoclonal anti-glutamic acid dec-arboxylase (Gad) 67 IgG$_{2a}$/Fab-A647 complexes] (Fig. 5b–j and

Supplementary Movie 4). We also performed other combinations of multicolor staining and imaging: 4-color staining with BOBO-1, rat monoclonal anti-Sst IgG$_{2b}$/Fab-Cy3, goat polyclonal anti-ChAT IgG/Fab-A594 and mouse monoclonal anti-Tph2 IgG$_1$/Fab-A647 complexes (Supplementary Fig. 8a and Supplementary Movie 4) and 3-color staining of an Alzheimer's disease model

**Fig. 4 CUBIC-HV is applicable to various antibodies and dyes. a** Results of whole-brain staining and LSFM imaging with various antibodies (yellow) and nuclear stains (SYTOX-G or BOBO-1, blue). The data were reconstructed with Imaris software. The staining, imaging and image processing conditions are summarized in Supplementary Data 2. Voxel size $8.3 \times 8.3 \times 9 \ \mu m^3$. L: left, R: right. Scale: 2 mm. **b** Nine sets of the whole-brain imaging data in **a** (stained with anti-phospho-Nf, calbindin D28K, PV, Sst, ChAT, Dbh, Th, Tph2, or Copeptin antibodies, respectively, with nuclear stains) were registered and aligned to the whole-brain images obtained with anti-NeuN antibody staining and nuclear staining (structural standard). All the transformed data were merged as a pseudo-multitarget whole-brain antibody staining image with Imaris software. The precise registration results are indicated by the brain regions merged across different datasets, such as the paraventricular nucleus (PVN) and the supraoptic nucleus (SON) (NeuN+/copeptin+), the locus coeruleus (LC) in the pons (Dbh+/Th+) and the Purkinje cell layer in the cerebellum (calbindin D28K+/PV+). L left, R right. Scale: 2 mm (whole-brain), 1 mm (reconstituted coronal sections), 0.5 mm (inset).

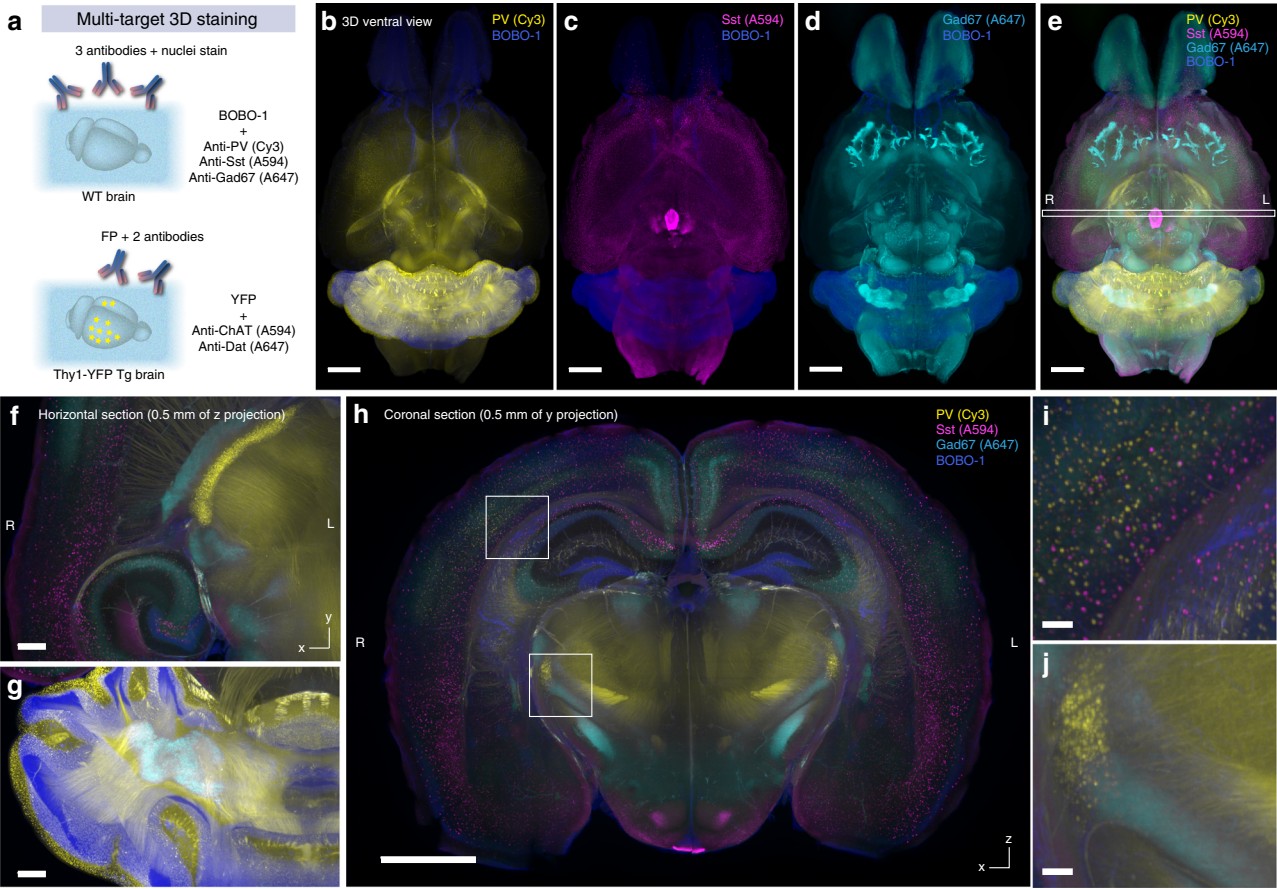

**Fig. 5 CUBIC-HV enables multicolor whole-organ staining and imaging. a** Experimental schematics of multichannel and multimodal whole-brain staining and imaging, shown in Figs. 5 and 6. **b–j** BOBO-1, PV (Cy3), Sst (A594), and Gad67 (A647) quadruple costaining images of the sole whole adult mouse brain reconstituted with Imaris software. (**b-d**) Individual antibody staining channels alongside the BOBO-1 channel. Voxel size $8.3 \times 8.3 \times 9 \ \mu m^3$. Scale: 2 mm. **e** Merged image of all four channels. Scale: 2 mm. **f, g** Enlarged horizontal (*x*–*y*) images of a representative section. Scale: 0.5 mm. **h** Reconstituted coronal (*x*–*z*) section image at the position indicated in **e**. Scale: 2 mm. **i, j** Enlarged images of the indicated positions in **h**. Scale: 0.1 mm.

mouse ($App^{NL-G-F}$ knock-in mouse)[41] with SYTOX-G, mouse monoclonal anti-β-amyloid $IgG_1$ (6E10)/Fab-A594 and mouse monoclonal anti-α-smooth muscle actin (SMA) $IgG_{2a}$/Fab-A647 complexes (Supplementary Fig. 8b). With an appropriate combination of host species and antibody isotypes, dyes, excitation lasers, and bandpass emission filters (see "Methods" for details), we could completely separate the signals from these different fluorescent channels.

One of the challenges of 3D staining and imaging is the combination of whole-organ staining with fluorescent protein (FP) labeling[8] (Fig. 5a). To test the compatibility of CUBIC-HV staining with FP labeling, we stained the brain of a Thy1-YFP-H transgenic mouse[42] with mouse monoclonal anti-dopamine transporter (Dat) $IgG_1$/Fab-A594 and goat polyclonal anti-ChAT IgG/Fab-A647 complexes. In the resulting images, the

YFP signal and immunolabeled Dat and ChAT signals were clearly observed at high intensity (Fig. 6a–g; Supplementary Movie 4). Two adjacent axonal projections (the Dat-labeled midbrain-striatum tract and the YFP-labeled corticospinal tract) were clearly distinguished in the 3D data (Fig. 6e–g). Taken together, the results allow us to conclude that CUBIC-HV is compatible with whole-organ multicolor and multimodal labeling and imaging.

**CUBIC-HV allows whole-organ cellular circuit analysis.** One application of 3D labeling and imaging is whole-brain neural circuit analysis with cellular resolution, which has been demonstrated in a limited manner by a few advanced reports[17,43,44]. We prepared a Gad2-Cre knock-in mouse brain[45] labeled with rabies

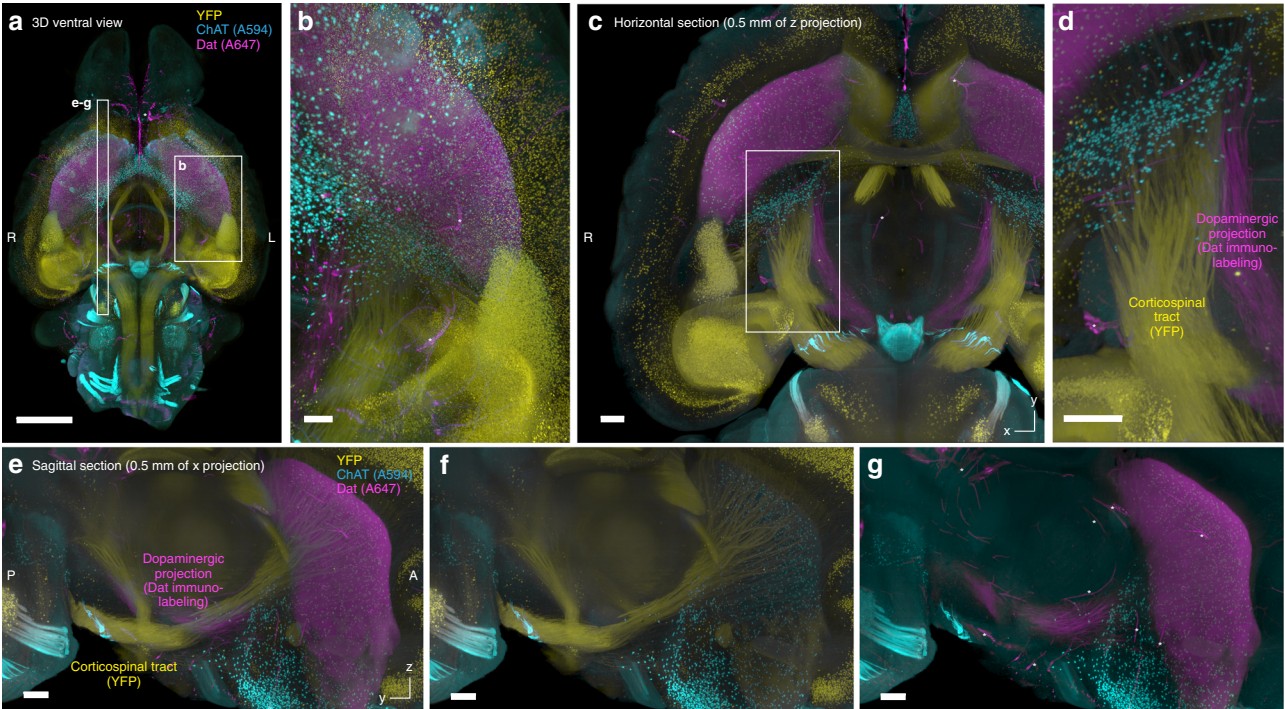

**Fig. 6 CUBIC-HV enables multimodal whole-organ labeling and imaging. a-g** ChAT (A594) and Dat (A647) double-staining images of the sole Thy1-YFP-H Tg whole adult mouse brain. Voxel size 8.3 × 8.3 × 9 μm³. **a** Merged image of all three channels. Scale: 2 mm. **b** Enlarged image at the position indicated in **a**. Scale: 0.5 mm. **c** Horizontal view of a representative section. Scale: 0.5 mm. **d** Enlarged image at the position indicated in **c**. Neighboring projection tracts labeled with YFP expression or anti-Dat immunolabeling were visualized. Scale: 0.5 mm. **e–g** Reconstituted sagittal sections at the position indicated in **a**, visualizing the neighboring projection tracts. Scale: 0.5 mm. Asterisks in **e–g** indicate nonspecific vascular signals due to insufficient perfusion of the sample.

virus (RV) tracing vectors[46] injected into M1 region of the cerebral cortex (Fig. 7a). We subsequently stained the delipidated whole brain with anti-Sst/Fab-Cy3 complex and far-red cell-impermeant nuclear stain RedDot™2 to collect cell-type and structural information, respectively. CUBIC-HV successfully visualized the virally labeled and immunostained cells in separate color channels (Fig. 7b, c). In the whole-brain image, we identified the distribution of RV-GFP-labeled cells around the injection site (Fig. 7d) and in regions away from the injection site, including the ipsilateral somatosensory area (Fig. 7e). Cells with all possible labeling combinations, such as GFP- and mCherry-positive cells (Sst-negative starter cells), GFP- and Cy3-positive cells (Sst-positive putative input cells) and GFP-, mCherry-, and Cy3-positive cells (Sst-positive starter cells), were identified in the reconstituted y–z section of the injection site (Fig. 7f–j). These results indicate that the neural circuitry and cell types could be visualized with single-cell resolution in the whole-brain multi-modal labeling data.

To further test whether CUBIC-HV could capture the whole-brain neuronal response under different experimental conditions, we then conducted whole-brain c-Fos staining of mice with or without MK-801 administration (Fig. 8a). Whole-brain c-Fos expression was successfully visualized (Fig. 8b, enhanced by a convolution filter in Fig. 8c). The reconstituted y–z sagittal sections show that c-Fos labeling could cover the entire brain area (Fig. 8d). MK-801 administration strongly induced c-Fos expression in several brain regions, including the olfactory system of the forebrain, the paraventricular nucleus of the thalamus, and several hypothalamic and hindbrain regions. In contrast, the number of c-Fos-positive cells was more pronounced in the hippocampal dentate gyrus of the MK-801 (−) brain than

in the MK-801 (+) brain, as demonstrated by the detection of Arc-dVenus transgene reporter in our previous studies[36,47] (Fig. 8d, insets). Therefore, we conclude that CUBIC-HV can enable whole-brain analysis of neuronal responses.

To test whether the immunolabeled cells could be computationally detected for further quantitative analyses, we prepared a cell detection algorithm. We first attempted to reconstitute the RV-traced brain data with the centroids of the labeled cells (Supplementary Fig. 9a). After optimizing the parameters, the algorithm could reasonably detect the centroids of the GFP-, mCherry- or Sst-labeled cells in the z-stack images around the injection site (Supplementary Fig. 9b). Furthermore, the multi-labeled cells in Fig. 7g–j were successfully recapitulated in the corresponding centroid images (Supplementary Fig. 9c).

We also tested the whole-brain detection of c-Fos-labeled cells by using the 3D maxima function according to the ClearMap algorithm[17] (Supplementary Fig. 10a). The local 3D maxima of c-Fos–positive cells with higher intensities than the nonspecific background signal were successfully identified (Supplementary Fig. 10b, c). These results indicate that the SBR and image resolution of the collected data are sufficient for the whole-brain quantitative analysis of cellular responses with single-cell resolution. Further improvement of the SBR in staining and the use of machine learning-based algorithms for computational detection may help increase this sensitivity.

**CUBIC-HV allows organ-level anatomical comparison.** We finally tested the concept of organ-level histological comparisons between species. We stained an adult marmoset brain hemisphere and an whole adult mouse brain with SYTOX-G and the same set of antibodies [anti-glial fibrillary acidic protein (GFAP)/Fab-

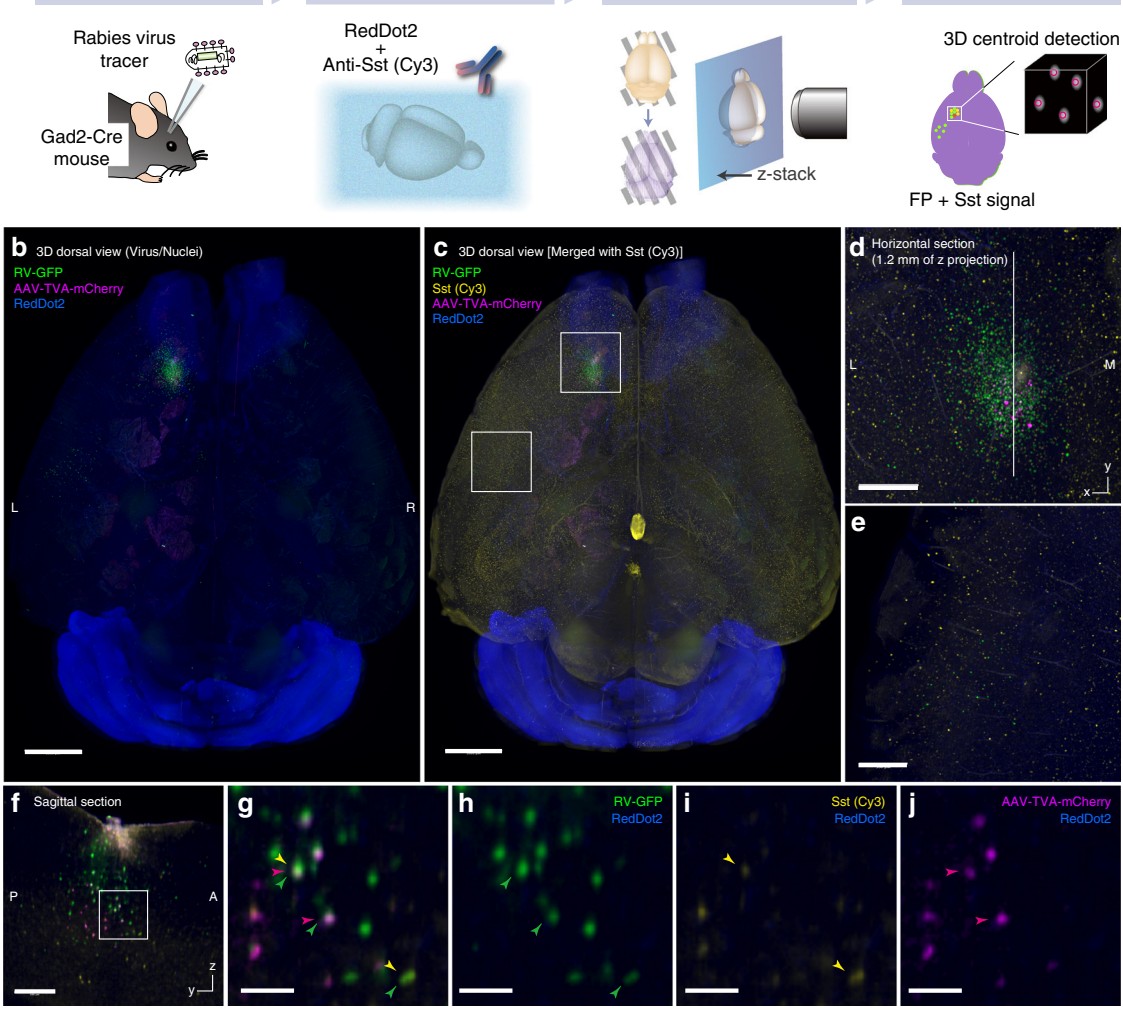

**Fig. 7 CUBIC-HV allows whole-organ cellular circuit analysis. a** Experimental schematic of whole-brain rabies virus (RV) tracing with cell-type immunolabeling. **b** Whole-brain image of the RV-injected Gad2-Cre adult mouse brain reconstituted with Imaris software. TVA-mCherry and RV-GFP were injected into left M1 region of the cerebral cortex. Voxel size 8.3 × 8.3 × 9 μm³. Scale: 2 mm. **c** Anti-Sst antibody staining channel was merged with the image in **b**. Scale: 2 mm. **d**, **e** Enlarged horizontal section images at the indicated positions in **c**, showing the RV-labeled neurons in the local and ipsilateral corticocortical circuits. Scale: 0.5 mm. **f** Reconstituted sagittal (y–z) section at the position indicated in **d**. Scale: 0.5 mm. **g-j** Cropped and enlarged images at the positions indicated in **f**. Starter neurons (green and magenta) and input neurons (green) with or without an Sst signal could be identified with single-cell resolution. Scale: 0.1 mm.

A594 and anti-α-SMA/Fab-A647 complexes] and compared the macroscopic distributions of GFAP-positive cells and α-SMA-positive arteries (Fig. 9a). Using CUBIC-HV, we successfully stained the whole adult marmoset brain hemisphere (Fig. 9b, c; Supplementary Movie 5). In the macroscopic brain images of both species, the strongest GFAP signal was identified in the surface region of the brain, a putative structure corresponding to the glia limitans (Fig. 9b, c). However, the GFAP-positive area was most prominent on the surface of the lateral ventricle and the pons of the marmoset brain, while in the mouse brain, it covered the surface of the hindbrain and the ventral surface of the hypothalamic regions. The observed differences between marmoset and mouse may reflect functional or regional differences in the structure between species.

A similar distribution pattern of the major arteries of the brains was observed in the middle cerebral artery (MCA), the posterior cerebral artery (PCA), the basilar artery (BA), and their peripheral branches in the brains (Fig. 9b). The reconstituted

sectional images further revealed that in both species, some of the branches of the MCA and the PCA supply the striatum and the hippocampus, respectively[48] (Fig. 9c). In addition, we successfully visualized the 3D distribution of perivascular astrocytes, which surround the corresponding arteries of both species very similarly (Fig. 9d). These data suggest evolutionary preservation of the overall vascularity of the brain.

We then tested the compatibility of CUBIC-HV with human specimens. We stained a postmortem cerebellum block (~1 cm³) with BOBO-1, Cy3-conjugated anti-GFAP antibody, and anti-α-SMA/Fab-A647 complex (Supplementary Fig. 11a). In the reconstituted sectional images, we compared the anatomy of the human cerebellum to the corresponding region of the marmoset and mouse cerebellum (Supplementary Fig. 11b–i). GFAP-positive Bergmann glial cells in the molecular layer (ML) and astrocytes associated with the vessels and white matter were evident in the human cerebellum (Supplementary Fig. 11d–e). In contrast, GFAP-positive astrocytes were more obvious in the

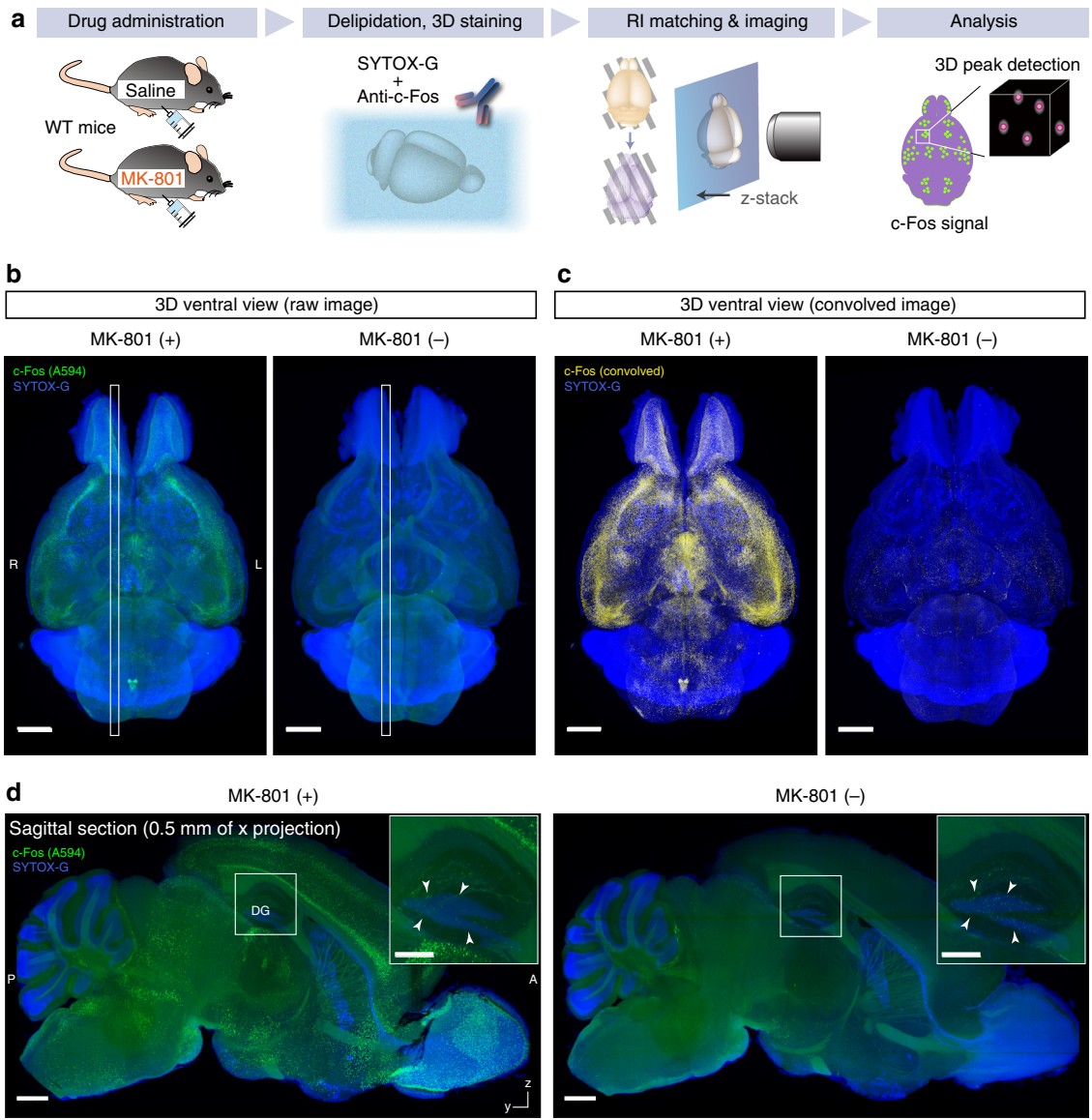

**Fig. 8 CUBIC-HV allows whole-organ cellular function analysis. a** Experimental schematic of whole-brain c-Fos antibody staining in different pharmacological conditions. **b** Whole-brain images of c-Fos-immunolabeled adult mouse brains with or without administration of NMDA receptor antagonist MK-801. The images were reconstituted with Imaris software. Voxel size $6.5 \times 6.5 \times 7 \mu m^3$. The white box indicates the position of the reconstituted sagittal image in **d**. Scale: 2 mm. **c** Resulting images following application of a convolution filter to the images in **b**. Scale: 2 mm. **d** Reconstituted sagittal images of brains in **b**, showing uniform labeling of c-Fos-expressing cells. The c-Fos-expressing cells of the dentate gyrus (DG) region were more obvious in the MK-801 (−) sample (arrowheads in insets). Scale: 1 mm (whole sagittal image) and 0.5 mm (insets).

granule cell layer (GCL) and the white matter of the marmoset cerebellum (Supplementary Fig. 11g). GFAP signals in the mouse cerebellum were generally weaker in these cerebellar layers and more pronounced in the perivascular areas and white matter. Since astrocytes play a critical role in cerebellar development[49], these distinct distributions of GFAP-positive astrocytes may account for some of the structural differences in the cerebellar layers among species. We confirmed the uniformity of 3D staining in the human and marmoset cerebellums by comparing the sectional images reconstituted from 3D data with the 2D-stained images of the physical section samples (Supplementary Fig. 11c, f, h).

These data indicate that the CUBIC-HV system could be extended to adult primate tissues, allowing for 3D histological comparisons among species. We summarize the concept of CUBIC-HV, including delipidation, staining, clearing, 3D imaging, and image analysis, in Fig. 9e.

## Discussion

We revealed that PFA-fixed and delipidated samples subjected to tissue clearing can be characterized as charged and cross-linked polypeptide gels (Fig. 1; Supplementary Fig. 1). The samples exhibited a swelling-shrinkage behavior under various chemical conditions, which resembled the patterns of fixed protein gels (further discussed in the Supplementary Discussion section). We exploited this insight to investigate essential experimental conditions for 3D staining under more simplified and controlled conditions by using fixed gelatin gel as a surrogate for the tissue (Fig. 2a; further discussed in Supplementary Discussion section). This insight is also pertinent to our CUBIC-X expansion microscopy approach[36]. Instead of using artificial water-absorbing polymers, we chemically enhanced the inherent gel properties of biological tissue in the CUBIC-X protocol. We also used gelatin gels for screening tissue-expanding chemicals. The current study further certifies the validity of our past strategy.

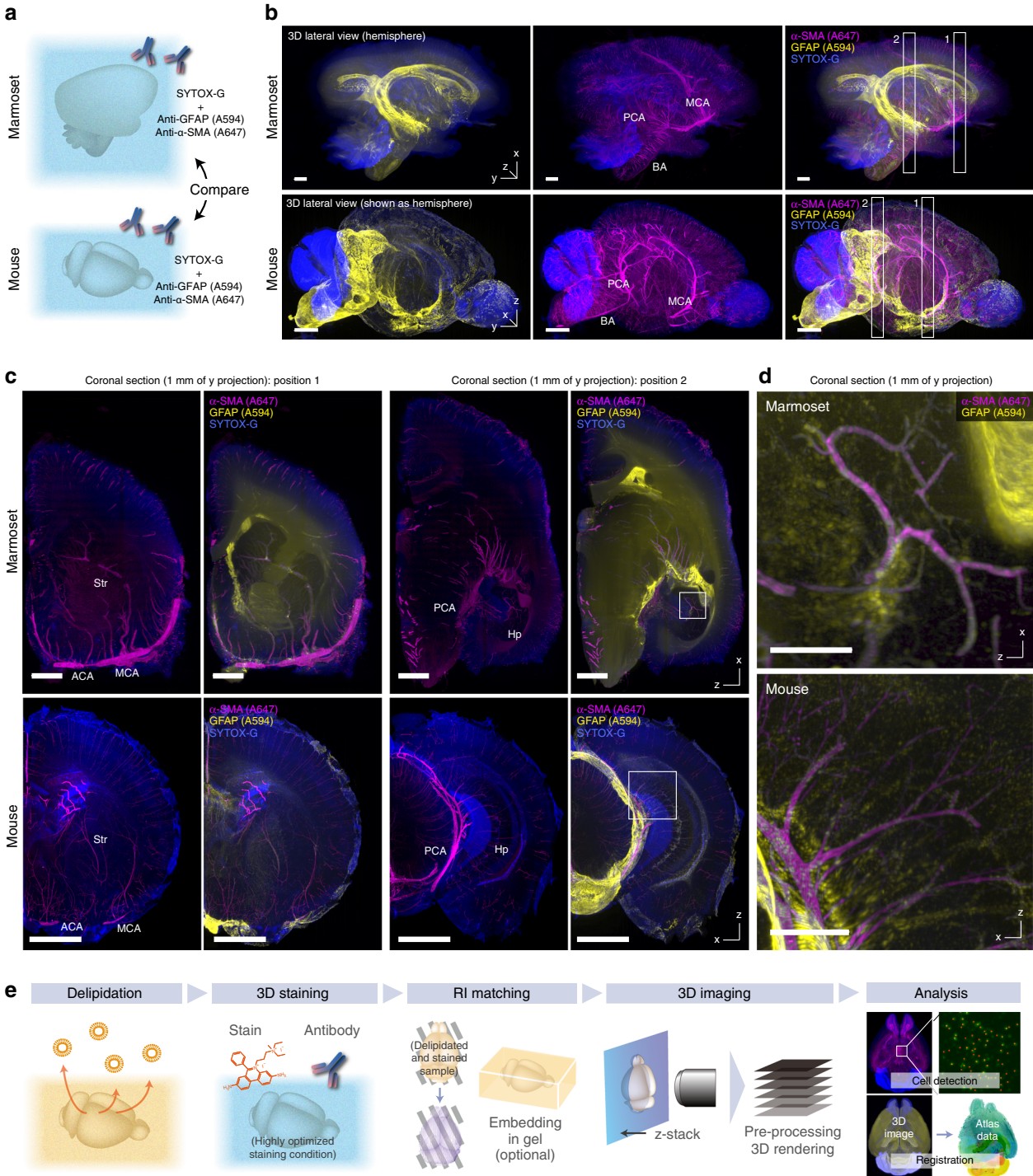

**Fig. 9 CUBIC-HV enables organ-level comparison of rodent and primate brains. a** Experimental schema of the whole-organ comparison analysis of rodent and primate brains. **b** Comparison of an adult marmoset brain hemisphere with the whole adult mouse brain (only the right hemisphere is depicted), costained with SYTOX-G, GFAP (A594), and α-SMA (A647) antibodies. Note that the orientations were different between samples. Images were reconstituted with Imaris software. Voxel sizes $16.4 \times 16.4 \times 16\ \mu m^3$ (for marmoset brain) and $8.3 \times 8.3 \times 9\ \mu m^3$ (for mouse brain). MCA middle cerebral artery, PCA posterior cerebral artery, BA basilar artery. Scale: 2 mm. **c** Reconstituted coronal sections of hemisphere images in **b**, showing the distributions of corresponding vessel structures and GFAP-positive cells. Note that the different orientations of the sections ($z$–$x$ and $x$–$z$ for the marmoset and mouse samples, respectively) could be compared because of the almost isotropic voxel resolution of the $z$-stack images. ACA anterior cerebral artery, Str striatum, Hp hippocampus. Scale: 2 mm. **d** Magnified images at the positions indicated in **c**, showing the similar distributions of vessel-associating GFAP-positive cells in the hippocampus. Voxel size in both images $8.3 \times 8.3 \times 9\ \mu m^3$. Scale: 0.5 mm. **e** Flowchart of staining-based whole-organ imaging and image analysis on CUBIC-HistoVIsion.

In principle, the bottom-up and objective approach conducted here is applicable to other 3D staining systems. Applying our optimized protocol to the existing 3D staining and clearing protocols AbScale and iDISCO+ markedly improved the staining results (Fig. 3h; Supplementary Fig. 4d). In this study, we did not test our protocol for hydrogel-tissue chemistry, which is adopted in techniques including CLARITY, PACT, and ExM[12,16,50]. However, a similar optimization approach to their 3D staining protocols[26,27] might be possible by considering the chemical properties of acrylamide gel (Supplementary Fig. 1g).

CUBIC-HV has the potential to push the boundaries of 3D histology. It can lead to discoveries of rarely reported cerebral astrocytic foci in physiological conditions (Supplementary Fig. 6h; further discussed in Supplementary Discussion), systematic analysis of whole-brain neural circuitry and function (Figs. 7 and 8), organ-level histological comparisons between species (Fig. 9), and 3D clinical pathology of postmortem human specimens (Supplementary Fig. 11). CUBIC-HV can also be applied to other organs based on the consistency of gel properties among tissue types (Supplementary Fig. 1h; Supplementary Fig. 7a). The advantages of CUBIC-HV have boosted these achievements. First, CUBIC technologies, including CUBIC-HV, are designed to be stable, easy, reproducible, and safe[13]. Second, CUBIC is compatible with FP reporters, allowing the simultaneous observation of genetically introduced fluorescent protein and staining signals (Figs. 6 and 7). Third, a highly optimized staining condition enables stable volumetric staining of whole organs and the whole body with a practically feasible number of stains and antibodies (e.g., 5 μL of 1 mg/mL NeuN antibody product for the whole mouse brain, see "Methods" for details) and without any specialized device, unlike techniques described in previous studies[26,27]. The signal intensity and SBR of the acquired images were sufficient for subsequent computational image processing and analysis. These results ease previous concerns regarding the applicability of CUBIC to staining experiments[51,52].

On the other hand, the current CUBIC-HV protocol still has potential limitations. Multiplex protein labeling or simultaneous detection of RNA tested in other studies[25,53] are challenging for our current protocol. The possible influence on subcellular structures and molecular integrity during clearing and staining should be considered, as noted in other studies[15,25,53], although we confirmed their preservation in our samples (Supplementary Fig. 6i–n; further discussed in Supplementary Discussion section). Issues related to thick primate tissues, such as the penetration efficiency of dyes/antibodies or a substantial absorption of light sheets in the sample (the so-called inner filter effect), should also be taken into account.

Whole-organ profiling of protein expression and post-transcriptional modification and localization, together with single-cell resolution resources[36], has been increasingly demanded in recent years. A superior 3D staining protocol would also facilitate 3D clinical pathology for more objective and accurate disease diagnoses[20–22]. The 3D histology realized by CUBIC-HV thus offers an ever-increasing opportunity for systemwide whole-organ and whole-body analysis with single-cell resolution in life science and medicine.

## Methods

**Tissue specimens**. For most of the staining experiments, we used 8-week-old male C57BL/6 N mice (Japan SLC). For the whole-brain c-Fos staining experiments (Fig. 8), 6-week-old C57BL/6 N mice were acclimated to our housing conditions (temperature range: 21–25 °C, humidity range: 40–70%, light cycle: on and off at 8 am and 8 pm) for two weeks. Then, we intraperitoneally administered 20 mg/kg MK-801 in saline (Sigma-Aldrich #M107) or an equal volume of saline at Zeitgeber Time (ZT) 2 and sacrificed the animals at ZT 4. We also used an 8-month-old female Thy1-YFP-H transgenic mouse[42] (Fig. 6). Furthermore, we introduced 3-week-old male $App^{NL-G-F}$ knock-in mice[41] (RBRC06344, provided by the RIKEN

BRC through the National BioResource Project of the MEXT/AMED, Japan) and housed them until they were 9 months old for further use (Supplementary Fig. 8). For the rabies virus labeling experiment (Fig. 7), an 8-week-old male Gad2-IRES-Cre knock-in mouse (Gad2$^{tm2(cre)Zjh}$/J, Jackson laboratory #010802)[45] was used. The animal was injected with 150 nL of AAV mixture (serotype 2 CAG-FLEx-TCb and CAG-FLEx-RG[54]) into the left primary motor cortex (1.2 mm lateral, 1.5 mm anterior from bregma, and 0.45 mm ventral from the brain surface) and was maintained under a 7 am–7 pm light-dark cycle. Two weeks later, 150 nL of EnvA pseudotyped RV-GFP ($1 \times 10^9$ infectious particles mL$^{-1}$, prepared following Osakada et al.[46]) was injected into the same location. The brain was sampled 4 days after RV injection.

For sampling the brains, the mice were sacrificed by an overdose of pentobarbital (>100 mg/kg, i.p.; Somnopentyl, Kyoritsu Seiyaku, or pentobarbital sodium salt; nacalai tesque #02095-04) and then transcardially perfused with 10 mL of PBS containing ~10 U/mL heparin followed by 20–30 mL of 4% PFA in PBS. Then, the brain was excised from the skull and postfixed with the same fixative for 8–24 h at 4 °C. These brains were used for the subsequent clearing procedure or stocked at 4 °C in PBS with 0.05% NaN$_3$ or −80 °C in O.C.T. compound until required for use.

For Fig. 2, we used an infant marmoset (postnatal day 1) with a low birth weight (22 g) who was neglected by its parents. The animal was sacrificed by an overdose of pentobarbital (>50 mg/kg, i.p.; Somnopentyl, Kyoritsu Seiyaku) after an intramuscular injection of ketamine (30 mg/kg; Ketalar, Daiichi Sankyo Propharma) and whole blood removal by transcardial perfusion with 20 mL of PBS containing ~10 U/mL heparin. Then, the animal was transcardially perfused with 150 mL of 4% PFA in PBS and 20 mL of ScaleCUBIC-1 (modified CB perfusion[14]). The skin was removed for the subsequent clearing procedure.

For Fig. 9, we used a fixed brain hemisphere derived from an adult marmoset (7 years old, female, 290 g body weight). For 2D-section staining (Supplementary Fig. 11h), we used a fixed cerebellum derived from another adult marmoset (6 years old, male, 304 g body weight). The animals were sacrificed by an overdose of pentobarbital (80–100 mg/kg pentobarbital, i.p.; Somnopentyl, Kyoritsu Seiyaku) and then transcardially perfused with 200 mL of PBS and 350 mL of 4% PFA in PB. The brain was excised from the skull and postfixed with the same fixative overnight at 4 °C. The brain was washed with PBS before clearing.

All experimental procedures and housing conditions of the animals were approved by the Animal Care and Use Committees of the Graduate School of Medicine and the Graduate School of Agricultural and Life Sciences of the University of Tokyo, RIKEN Kobe Institute, National Institute for Physiological Sciences, and the Graduate School of Medicine of Kyoto University. All of the animals were cared for and treated humanely in accordance with the Institutional Guidelines and with the recommendations of the United States National Institutes of Health for experiments using animals.

For Supplementary Fig. 11, we prepared ~1 cm$^3$ blocks of human cerebellum derived from an amyotrophic lateral sclerosis patient who underwent postmortem pathological dissection at Niigata University Hospital after his death (male, 73 years old, died of respiratory failure). No histopathological alterations were observed in the cerebellum. The brain had been fixed in 20% buffered formalin for 4 weeks before clearing. For the Klüver–Barrera (K–B) staining and 2D-section staining experiments (Supplementary Fig. 11c, e, f), additional cerebellum parts were dissected after 3 years in stock formalin. The study was approved by the Ethical Review Boards of the Brain Research Institute, Niigata University (No. 1992) and the Graduate School of Medicine of The University of Tokyo (No. 10544 and No. 10714) and was performed in accordance with the committee guidelines and regulations. Informed consent was obtained from the next of kin of the patient.

**Dyes and dye-labeled secondary antibodies**. In this study, we used the following dyes and secondary antibodies: propidium iodide (PI, Molecular Probes, #P21493), SYTO$^{TM}$ 16 (designated SYTO 16, Thermo Fisher Scientific #S7578), DAPI (Dojindo Molecular Technologies #D0523), SYTOX$^{TM}$ Green (designated SYTOX-G, Thermo Fisher Scientific #S7020), BOBO$^{TM}$-1 Iodide (462/481) (designated BOBO-1, Thermo Fisher Scientific #B3582), RedDot$^{TM}$2 Far-Red Nuclear Stain (designated RedDot2, Biotium #40061), NeuroTrace$^{TM}$ 640/660 Deep-Red Fluorescent Nissl Stain (Thermo Fisher Scientific #N21483), eosin (Acid Red 87, Tokyo Chemical Industry/TCI #T0037), anti-NeuN (A488-conjugated, Merck Millipore MAB377X), anti-mouse secondary IgG (Thermo Fisher Scientific, #A11029 for Alexa 488, #A21203 for Alexa 594), anti-rabbit secondary IgG (Thermo Fisher Scientific #A21206 for Alexa 488, #A10040 for Alexa 546, #A21207 for Alexa 594, #A31573 for Alexa 647), and FabuLight$^{TM}$ AffiniPure Fc specific Fab Fragments [Jackson ImmunoResearch laboratories; anti-mouse IgG$_1$ (#115-547-185 for Alexa 488, #115-167-185 for Cy3, #115-587-185 for Alexa 594, #115-607-185 for Alexa 647), anti-mouse IgG$_{2a}$ (#115-587-186 for Alexa 594, #115-607-186 for Alexa 647), anti-mouse IgG$_{2b}$ (#115-587-187 for Alexa 594), anti-rat IgG$_1$ (#112-167-008 for Cy3, #112-587-008 for Alexa 594), anti-rabbit IgG (#111-167-008 for Cy3, #111-587-008 for Alexa 594, #111-607-008 for Alexa 647), and anti-goat IgG (#805-587-008 for Alexa 594)]. The primary antibody and secondary Fab complexes were prepared by mixing the constituent compounds at a 1:0.5 ~ 1:1 weight ratio and incubating for at least 1 h at room temperature. Then, the complex was added to the staining buffer.

**Preparation of optically cleared tissue with the CUBIC protocol**. Adult mouse brains were delipidated with Sca*le*CUBIC-1A [10 wt% of Triton X-100 (nacalai tesque #12967-45), 5 wt% of N,N,N',N'-Tetrakis(2-hydroxypropyl)ethylenediamine (Quadrol, TCI #T0781), 10 wt% of urea (nacalai tesque #35904-45), and 25 mM NaCl (nacalai tesque #31319-45) in water], Sca*le*CUBIC-1 [15 wt% of Triton X-100, 25 wt% of Quadrol, and 25 wt% of urea in water] or CUBIC-L [10 wt% of N-butyldiethanolamine (TCI #B0725) and 10 wt% of Triton X-100 in water], according to our published procedures[13,40] (http://cubic.riken.jp). We typically treated the whole mouse brain samples for 10 days with Sca*le*CUBIC-1A or 3-4 days with CUBIC-L. In the case of *App*NL-G-F mouse brains, the delipidation period was increased to 7 days. The procedure with Sca*le*CUBIC-1 is described below (see the "Quantification of proteins and lipids in brain samples" section). After delipidation, the brains were intensively washed with PBS and stored in PBS with 0.05% NaN₃ at 4 °C until required for use. For imaging purposes, delipidated and stained mouse brains were RI-matched with CUBIC-R+(N) [45 wt% of antipyrine (TCI #D1876) and 30 wt% of nicotinamide (TCI #N0078) in water, buffered with 0.5% (v/w) N-butyldiethanolamine (pH ~10), (N) represents nicotinamide] or CUBIC-R+(M) [the corresponding reagent of CUBIC-RA in Tainaka et al.[40]: 45 wt % of antipyrine and 30 wt% of N-methylnicotinamide (TCI #M0374) in water, buffered with 0.5% (v/v) N-butyldiethanolamine (pH ~10), (M) represents N-methylnicotinamide]. The sample was first immersed in 1:1 water-diluted CUBIC-R+(N) or CUBIC-R+(M) at room temperature for 1 day. Then, the sample was immersed in nondiluted CUBIC-R+(N) or CUBIC-R+(M) at room temperature for 2-3 days. The sample was finally embedded in 2% (w/v) agarose gel prepared with CUBIC-R+(N) or CUBIC-R+(M) according to our published procedure[36,40]. CUBIC-R+(M) was used for the preservation of fluorescent protein signals in the Thy1-YFP-H Tg brain and the RV-labeled brain (Figs. 6 and 7).

For clearing and PI staining of the infant marmoset body (Fig. 2), delipidation and decolorization after fixation, CB perfusion and skin removal was continued in Sca*le*CUBIC-1 with shaking for 2.5 months at 37 °C. The reagents were replaced every 2–3 days during the first month and then every week during the remaining period. The sample was further preserved in Sca*le*CUBIC-1 at room temperature until it was used for the PI staining and imaging experiment. After washing in PBS, the sample was stained with PI (10 µg/mL) in 0.1 M PB containing 1.5 M NaCl for 7 days at 37 °C, washed with PBS and 10 wt% of imidazole (for NaCl removal), and then RI-matched with CUBIC-R+(N). The sample was first immersed in 1:1 water-diluted CUBIC-R+(N) at room temperature for 4 days. Then, the sample was immersed in nondiluted CUBIC-R+(N) at room temperature for 6 days. The CUBIC-R+(N) was refreshed every 2 days.

The adult marmoset brain hemisphere (Fig. 9) was immersed in CUBIC-L with shaking for 17 days at 37 °C and then for 6 days at 45 °C. The CUBIC-L was refreshed every 1–2 days during the delipidation. After 3D staining, the sample was first immersed in 1:1 water-diluted CUBIC-R+(N) at room temperature for 1 day. Then, the sample was immersed in nondiluted CUBIC-R+(N) at room temperature for 5 days. The sample was finally embedded in 2% (w/v) agarose gel prepared with CUBIC-R+(N). For 2D-section staining (Supplementary Fig. 11h), a piece of dissected cerebellum (~5 mm thick) was treated with CUBIC-L for 14 days at 37 °C followed by immersion in 40% sucrose/PBS and cryosection.

The delipidation of the human postmortem cerebellum block (Supplementary Fig. 11) had been completed before we developed CUBIC-L. Thus, the specimen was dissected to a size of ~1 cm³ and immersed in Sca*le*CUBIC-1A with shaking for 9 weeks at 37 °C. The reagent was refreshed every 1–2 days during the delipidation. After washing in PBS, the brain hemisphere was stored in nonfrozen stock solution (0.1 M PB, 40% glycerol and 40% ethylene glycol) at −30 °C until use. Before 3D staining, the stock solution was washed with PBS, and the sample was further delipidated with CUBIC-L for an additional 3 days at 37 °C. After 3D staining, the sample was first immersed in 1:1 water-diluted CUBIC-R+(N) at room temperature for 1 day. Then, the sample was immersed in nondiluted CUBIC-R+(N) at room temperature for 3 days. The sample was finally embedded in 2% (w/v) agarose gel prepared with CUBIC-R+(N). For 2D-section staining (Supplementary Fig. 11c, f), a piece of dissected cerebellum (2–3 mm thick) was treated with Sca*le*CUBIC-1A for 7 days at 37 °C followed by immersion in 40% sucrose/PBS and cryosection.

**Quantification of proteins and lipids in brain samples**. PFA-fixed mouse brains (ICR, 7 weeks old, male) were washed with PBS and immersed in 1:1 water-diluted Sca*le*CUBIC-1 at 37 °C for 6 h with gentle shaking and then immersed in Sca*le*-CUBIC-1 at 37 °C overnight with gentle shaking. The Sca*le*CUBIC-1 was refreshed every day throughout the delipidation. Meanwhile, control samples were incubated in PBS at 37 °C with gentle shaking without solvent exchange. The day 0–5 brain samples were then subjected to protein, lipid, and water quantification. To quantify the residual protein and lipid content, the samples were thoroughly crushed and then sonicated in 1% Triton X-100 in PBS to obtain a homogenized suspension. After adjusting for the total weight of each sample, the protein was quantified with a Microplate BCA™ Protein Assay Kit–Reducing Agent Compatible (Thermo Fisher Scientific #23252) and OD570 according to the instruction manual. The phospholipid and cholesterol contents were quantified with LabAssay™ Phospholipid (FUJIFILM Wako #296-63801) and LabAssay™ Cholesterol (FUJIFILM Wako #296-65801), respectively. The water content of the brain sample was determined by measuring the constant weight lost after drying in a convection oven

(WFO-1001SD, EYELA) at 105 °C for 24–48 h, according to previous studies[55,56]. The transmittance of the brain samples was quantified with a Spectral Haze Meter SH 7000 (Nippon Denshoku Industries) according to our previous study[14].

**Measurement of SAXS of brain samples**. For the SAXS experiments (Fig. 1; Supplementary Fig. 1), we prepared 2 mm-thick coronal sections of the Sca*le*-CUBIC-1A-delipidated mouse brains. The sections were immersed in PB containing various concentrations of NaCl and then embedded in Kapton films.

The SAXS experiment for the samples was performed at the Frontier Soft Matter Beamline (FSBL; BL03XU) of SPring-8 (Harima, Japan). The detector for SAXS was a Pilatus3 1 M (DECTRIS), which was placed 4 m behind the sample. X-rays with a wavelength of 1.0 Å and a spot size of 60 × 200 µm were used. The X-ray beam irradiated the cortical region of the coronal sections for 1–2 s at each position.

The other SAXS experiment was performed at the BL10C beam station in the Photon Factory of KEK (Tukuba, Japan). The SAXS detector was a Pilatus3 2 M (DECTRIS), which was placed 2 m behind the sample. X-rays with a wavelength of 1.0 Å and a spot size of 180 × 600 µm were used. The X-ray beam irradiated the cortical region of the coronal sections for 60 s at each position.

The obtained 2D scattering profiles were converted to 1D scattering profiles as a function of the magnitude of the scattering vector (**q**) with Igor Pro 6.3-based custom-made software by circularly averaging the data around the beam center. For 2D data with an anisotropic scattering profile, simple line averaging at a given image angle was performed instead of circular averaging. The background scattering from the air and a sample cell (Kapton films) was subtracted from the raw scattering profiles.

**Swelling-shrinkage assay of the gels**. For the swelling-shrinkage assays of gels (Fig. 1; and Supplementary Fig. 1), we prepared cerebral hemispheres delipidated with Sca*le*CUBIC-1A, fixed type-B gelatin gels (nacalai tesque #16631-05, 5% (w/v), fixed with 4% PFA), fixed agarose gels (Agarose-S, FUJIFILM Wako #318-01195, 0.5% (w/v), fixed with 4% PFA) and polyacrylamide gels [40% acrylamide/Bis 29:1 mix, nacalai tesque #06141-35, diluted to 8% (v/v) and polymerized with the addition of 0.1% ammonium persulfate (nacalai tesque #02601-25) and 0.1% TEMED (Sigma-Aldrich #T9281)]. These artificial gels were shaped into discoidal gels by using the lid of a 96-well plate. For the assays, the prepared gels were first washed in water for 1 day at room temperature and then treated with a series of solvents [fractions of 0.5 to 500 mM NaCl, 0.6 M triethanolamine (TEA, FUJIFILM Wako #145-05605, adjusted to pH 10.5), a mixture of 0.1 M HEPES (nacalai tesque #17514-15) and 0.1 M TEA (adjusted to pH 9.5–6.5), and a mixture of 0.1 M MES (DOJINDO #349-01623) and 0.1 M TEA (adjusted to pH 5.5), 70 µM or 0.4 mM p-toluenesulfonic acid monohydrate (TCI #T0267, adjusted to pH 4.5 or 3.5, respectively) and fractions of 0 to 80% v/v of diluted acetone (nacalai tesque #00317-25) in water] at room temperature every day.

Gel images were captured by a Bio-Lad ChemiDoc XRS+ system and the included Image Lab software (version 4.1). We acquired bright-field images (for the delipidated brain) or autofluorescence images under UV illumination (for the artificial gels) every day before the solvent exchange. The captured images were processed with Fiji/ImageJ (http://fiji.sc/Fiji). A binary image was prepared with the "Auto threshold" function of the software and some manual adjustment to detect the gel region. Then, the area was calculated with the "Analyze particle" function. The gel size (S/S0) in each solution was calculated by dividing the area (S) of each gel by the mean area of the gels in PBS solution (S0, $n = 8$).

The resulting swelling-shrinkage curves were not necessarily consistent with those of past reports[30,57] because the patterns can be affected by the concentration of polymer, cross-linking methods, incubation time for gelation, or the type of substituted residue.

**Numerical simulation of 3D staining patterns inside the gel**. Numerical integration of systems of partial differential equations was carried out by using Mathematica software version 11.3 (Wolfram Research) with the built-in function "NDSolveValue".

The reaction-diffusion process of the reversible binding and unbinding scheme of the reactive solute (i.e., dyes and antibodies) and its binding target in 2D space is described below:

$$\frac{\partial[S]}{\partial t} = D_{\text{diff}} \cdot \left( \frac{\partial^2[S]}{\partial x^2} + \frac{\partial^2[S]}{\partial y^2} \right) + k_{\text{off}}[ST] - k_{\text{on}}[S][T], \tag{4}$$

$$\frac{\partial[T]}{\partial t} = k_{\text{off}}[ST] - k_{\text{on}}[S][T], \tag{5}$$

$$\frac{\partial[ST]}{\partial t} = -k_{\text{off}}[ST] + k_{\text{on}}[S][T]. \tag{6}$$

[S], [T] and [ST] denote the concentrations of the reactive solute, the binding target and the solute-target complex, respectively. $D_{\text{diff}}$, $k_{\text{on}}$, and $k_{\text{off}}$ denote the diffusion coefficient, the binding rate, and the unbinding rate, respectively. The simulation space was set as a 20 × 20 square. A circle with a 1.0 radius, namely, the gel space, was placed at the center of the square. The simulation was started with the following parameters and initial conditions unless otherwise stated: $D_{\text{diff}} = 1$;

$k_{on} = 1$, $k_{off} = 0.01$; initial $[S]$ outside of the circle = 100; initial $[S]$ within the circle = 0; initial $[T]$ outside of the circle = 0; initial $[T]$ within the circle = 100; initial $[ST]$ for the entire simulation space = 0. The numerical integration was continued until time = 1. All parameters and time scales were unitless and chosen such that the simulation with initial default values listed above resulted in an almost homogenous distribution of $[ST]$ inside the circle at time = 1. The time step of numerical integration was determined by the build-in Mathematica function "NDSloveValue", with the option MaxStepFraction = 0.003. This option constrains the maximum fraction of the total simulation range used for numerical integration.

Note that the binding target and the solute-target complex ($[T]$ and $[ST]$) do not diffuse outside of the circular gel. All numbers in this simulation are unitless because the goal of the simulation was not to elucidate the exact physical parameters needed to recapitulate the experimental results accurately.

**Experimental evaluation of 3D staining conditions.** For Figs. 2 and 3, 5% (w/v) type-B gelatin gel (nacalai tesque #16631-05) mixed with 100 µg/mL salmon testis DNA (Sigma-Aldrich #D1626) or 0.1-0.5 µg/mL rabbit immunoglobulin (Sigma-Aldrich #I5006) was dissolved in water containing 0.5% PFA. The gelatin solution was poured into the holes of a Teflon plate (H 5 mm × φ 5 mm), incubated for gelation for at least 2 h at 4 °C and room temperature, and then fixed with 4% PFA in PBS overnight at 4 °C.

The DNA-containing gelatin gels were 3D stained with PI (10 µg/mL) or SYTO 16 (1:100) for 3 h or 6 h, respectively, at 32 °C in either 10 mM HEPES (pH 7.5) or ScaleCUBIC-1A (0.25× diluted) containing NaCl. Plain gelatin gels were also prepared and stained with eosin (0.2%) in the same buffers for 1 h. The buffers were supplied with 20% sucrose for the gels to be cryosectioned immediately after staining.

The rabbit immunoglobulin-containing gelatin gels were 3D stained in HEPES-TSC buffer [10 mM HEPES (pH 7.5), containing 0.1% (v/v) Triton X-100, 200 mM NaCl, and 0.5% (w/v) casein (FUJIFILM Wako #030-01505)] containing anti-rabbit secondary IgG (A594) (10 µg/mL) for 48 h at 32 °C. Then, the gels were briefly washed in PBS, immersed in 30% sucrose in PBS, and subjected to cryosection. The concentrations of rabbit immunoglobulin and gelatin in the gel, the concentration of secondary IgG, NaCl and Triton X-100 in the buffer, the type of Alexa Fluor dye of the secondary IgG, or the staining temperature were occasionally changed to evaluate the parameters (Fig. 3, Supplementary Fig. 3).

Delipidated cerebellar hemispheres were 3D stained with PI (2 µg/mL), SYTO 16 (1:150) for 24 h, DAPI (2.5 µg/mL) for 40 h, SYTOX-G (1:2500) for 40 h, RedDot2 (1:150) for 23 h or NeuroTrace Deep-Red (1:100) for 24 h at 32 °C in either 10 mM HEPES (pH 7.5) or ScaleCUBIC-1A containing NaCl. Delipidated cerebral hemispheres were 3D stained with eosin (2%) for 4 h at 32 °C in the same buffers. Then, the samples were washed with PBS, immersed in 40% sucrose in PBS, and subjected to cryosection. For post-2D staining, the sections were further stained with DAPI (1:500) in PBS for 30 min at room temperature before observation.

Delipidated cerebellar hemispheres were also used for 3D immunostaining. For enzyme digestion (Fig. 3e), the delipidated samples were digested under stringent conditions with collagenase-P (Sigma-Aldrich #11213857001, 1 mg/mL) in carbonate buffer [mixture of 50 mM sodium carbonate (nacalai tesque #31310-35) and 50 mM sodium hydrogen carbonate (nacalai tesque #31213-15) supplied with 150 mM NaCl and 25-100 µM EDTA, adjusted to pH 10] or hyaluronidase (Sigma-Aldrich #H4272 or #H3884, 3 mg/mL) in CAPSO buffer [10 mM CAPSO (Sigma-Aldrich #C2278) and 150 mM NaCl, adjusted to pH 10] for 18 h at 37 °C before 3D staining. For the 1-step staining (Fig. 3c, e), the delipidated samples were 3D stained with either primary antibodies [A488-conjugated mouse anti-NeuN (Merck Millipore #MAB377X, 10 µg/mL), a complex of mouse anti-NeuN (Merck Millipore #MAB377, 10 µg/mL) and anti-mouse secondary Fab (A488) (1:1 as the weight ratio), or mouse anti-NeuN (10 µg/mL) alone] in HEPES-TSC buffer [10 mM HEPES (pH 7.5), 10% (v/v) Triton X-100, 200 mM NaCl, and 0.5% (w/v) casein] for 3–5 days at 32 °C. After 3D staining, these samples were washed and immersed in 40% sucrose in PBS and subjected to cryosection for observation. For the 2-step staining (Fig. 3d), the mouse anti-NeuN-stained sample was subsequently 3D stained with anti-mouse secondary IgG (A488) (10 µg/mL) in HEPES-TSC buffer for 5 days at 32 °C. After 3D staining, these samples were washed and immersed in 40% sucrose in PBS and subjected to cryosection for post 2D staining. The sections were briefly fixed with 4% PFA for 30 min at room temperature and further stained with rabbit anti-NeuN primary antibody (Merck Millipore ABN78, 1:500) in PBST [0.1% (v/v) Triton X-100] for 2 h at room temperature followed by anti-rabbit secondary IgG (A594) (2 µg/mL) in PBST for 1 h at room temperature or stained only with anti-mouse secondary IgG (A594) (2 µg/mL) in PBST for 1 h at room temperature.

The cryosections (a 200 µm thick portion of the gelatin gel, 2–3 mm from the top, or a 50 µm thick section of the cerebellar hemisphere, ~2 mm from the lateral edge) were prepared and used for post-2D staining and imaging with an upright fluorescence microscope (BX51, Olympus) with a 4× objective lens (details below). Quantification of the signal intensity inside the gelatin gel was performed as shown in Fig. 2a. The images were processed with Fiji/ImageJ functions to remove outliers (radius = 10, threshold = 50), subtract background signals (background intensity manually determined with a nonstained gel image), crop the gel region and adjust its size (width = 1500 pixels). Then, the intensity profiles along the gel diameter

(1500 × 30 pixels along 6 diameters) were calculated. The final profile graph was prepared with Excel software as the mean intensities ± SDs of these 6 diameter profiles.

**Antibody selection for 3D staining.** We summarize all the antibodies and their dilution ratios in Supplementary Data 1 and 2. We started by testing the antigenicity of the antibodies (Supplementary Data 1) on CUBIC-delipidated mouse brain sections. Among the 44 tested antibodies, 42 (95%) resulted in an expected staining pattern on the CUBIC-L-delipidated sections (Supplementary Fig. 5; Supplementary Data 1). We occasionally tested multiple antibodies for the same target [glutamic acid decarboxylase (Gad 65/67, dopamine transporter (DAT), choline acetyltransferase (ChAT), PKCα, glial fibrillary acidic protein (GFAP), oligodendrocyte transcription factor 2 (Olig2), microtubule-associated protein 2 (MAP2), c-Fos or β-amyloid]. However, the signal intensity or SBR was more dependent on the antibody itself than on the targeted molecule (Supplementary Data 1), suggesting that the different staining results (Supplementary Fig. 5) might be more attributable to the detection ability and clonality of the antibody rather than a decrease in the protein contents.

Then, we chose 40 antibodies (including mouse anti-NeuN in Fig. 3h, i) for the 3D staining tests of the delipidated hemisphere or whole brain (Supplementary Data 1). In this step, we checked their signal intensities, SBRs, compatibilities with enzymes and additives, and penetration efficiencies during a reasonable period (1-2 weeks). We further excluded 15 antibodies due to low penetration efficiency, low SBR, or high nonspecific signals. Ten of them were polyclonal antibodies, implying a possibility that a mixture of antibodies for different epitopes might not give a sufficient initial concentration for each epitope, which might result in lower penetration efficiency or SBR. The remaining five were monoclonal antibodies. Two of them target a neurofilament or spine protein, suggesting that richer antigens or stronger binding perturbed their adequate penetration into the sample (Fig. 3a). The low concentration (0.03 mg/mL) of the neurofilament antibody product might have also accounted for the insufficient result. One of the anti-β-amyloid antibodies (82E1) was also categorized as having low penetration due to its too-strong affinity. A monoclonal anti-c-Fos antibody was excluded for further use due to nonspecific signals in the vessels when used for 3D staining. As a result, we finally selected 25 out of 40 (62.5%) as proper antibodies for the following whole-organ staining according to their macroscopic staining patterns, penetration efficiencies, and sufficient signal intensities.

For antibody selection in the experiments depicted in Fig. 9 and Supplementary Fig. 11, we adopted the criteria of (1) compatibility for the three different species, (2) expectation of preserved or varied patterns of labeled structures for comparison, (3) constraint in the multilabeling combination of IgG hosts and subtypes, and (4) sufficient signal intensity and penetration efficiency based on the results with mouse brains. We chose the combination of anti-GFAP and α-SMA antibodies to meet the criteria.

**3D staining by CUBIC-HistoVIsion1.0.** We provide a step-by-step protocol of 3D staining by CUBIC-HistoVIsion1.0 in the Supplementary Methods section. In brief, the CUBIC-delipidated tissue was stained with one of the proper nuclear stains [SYTOX-G (1:2500), BOBO-1 (1:400), RedDot2 (1:150)] in 4 mL of ScaleCUBIC-1A containing 500 mM NaCl for 3-5 days at 37 °C. If needed, the nuclear-labeled sample was then digested under stringent conditions with 3 mg/mL hyaluronidase in CAPSO buffer (pH 10) or 1 mg/mL collagenase-P in carbonate buffer (pH 10) for 24 h at 37 °C. Although we incorporated the step into the current protocol to demonstrate the digestion concept, it is considered optional due to possible overdigestion and loss of antigenicity. Indeed, this step was occasionally skipped for some antibodies because their antigens were incompatible with the treatment (Supplementary Data 1). Then, the sample was 3D stained with a dye-conjugated primary antibody or a complex of primary antibody and dye-conjugated secondary Fab fragment (1:0.5–1:1 weight ratio) in HEPES-TSC buffer [10 mM HEPES buffer, pH 7.5, containing 10% (v/v) Triton X-100, 200 mM NaCl, and 0.5% (w/v) casein] at 25, 32, or 37 °C for an appropriate duration (Supplementary Data 1). The buffer was occasionally supplied with Quadrol [by diluting 50 wt% of stock to the final 2.5 to 5%] and urea (0.5 to 2 M). The sample was immersed in a limited volume of staining buffer (e.g., 500 µL for a whole mouse brain in a 15 mL tube, Sarstedt #60.732.001) and shaken at 40-50 rpm during the staining; the slow shaking speed was critical for preserving the fragile sample. We also confirmed that higher shaking speeds did not promote faster penetration of the antibody. For some antibodies, we additionally incubated the staining tube at 4 °C for 1-5 days to stabilize the signals. After staining, the sample was briefly washed with 0.1 M PB with 10% Triton X-100 followed by PB without Triton X-100 for 2 h in total at the antibody reaction temperature. Then, the sample was postfixed with 1% formaldehyde (nacalai tesque #16222-65) in 0.1 M PB for 5–24 h at 25 °C. The sample was washed with PBS and used for RI matching with CUBIC-R+.

During the procedure, adequate care for the fragile samples was employed. We tended to use a metal spoon to handle them and avoided placing them directly onto a paper towel or tissue paper because the sample would become slightly sticky.

Some 3D staining tests [Supplementary Data 1, 3D staining result (with hemisphere) column] were performed with HEPES-TAC buffer (10 mM HEPES, 5% Triton X-100, 200 mM arginine-HCl (nacalai tesque #03323-32), 0.5% casein, final pH = 8.0) according to our preliminary staining protocol.

To evaluate the performance of our final 3D staining protocol, the 3D- and post-2D–staining results in the same sample were compared (Supplementary Fig. 4b, c) as was performed in a previous report[15]. We stained the CUBIC-L-delipidated whole adult mouse brains with the cell-impermeant nuclear stain SYTOX-G (1:2500 in 4 mL of ScaleCUBIC-1A, 500 mM NaCl) for 5 days at 37 °C with gentle rotation (3D). The samples were horizontally cryosectioned and restained with DAPI (2 μg/mL) in PBS for 2 h at room temperature (post-2D). As the control for uniform staining, unstained horizontal cryosections were also prepared and either used for SYTOX-G homogeneous staining (2D) (1:10000 in PBST for 1 h at room temperature) or left unstained, after which both sets of cryosections were stained with DAPI (post-2D). Similar staining procedures were applied for evaluating the 3D immunostaining procedure. CUBIC-L-delipidated and hyaluronidase-treated whole adult mouse brains were 3D stained with a complex of mouse anti-NeuN antibody (10 μg/mL) and secondary Fab (A594) fragment (1:1 weight ratio) in HEPES-TSC buffer supplied with 2.5% Quadrol. After 3D staining, 2D cryosections were prepared, refixed, and restained with rabbit anti-NeuN antibody (1:1000 in PBST) for 2 h at room temperature, followed by anti-rabbit and anti-mouse secondary antibodies (A488 and A594, respectively, 1:1000 each in PBST) (post-2D). In the last step, anti-mouse secondary antibody (A594) was simultaneously used to enhance the mouse anti-NeuN antibody signals because the secondary Fab was not fully retained during the post-2D staining. As the control for uniform staining, a nonstained brain was first cryosectioned and then stained with a complex of mouse anti-NeuN antibody (1 μg/mL) and secondary Fab (A594) (1:0.5 weight ratio) in PBST overnight at room temperature (2D). The sample was refixed and further stained with rabbit anti-NeuN antibody followed by anti-rabbit and anti-mouse secondary antibodies (A488 and A594, respectively) (post 2D). The staining efficiency was evaluated by comparing (1) the intensity profile in the middle portion of the section and (2) the colocalization parameter (Pearson's R value) between the pre- and post-staining channels at the exterior (ROI-1) and interior (ROI-2) regions of the section (Supplementary Fig. 4b, c).

AbScale-[15], iDISCO+-[17] (version Dec. 2016, https://idisco.info/idisco-protocol/), and SWITCH-mediated antibody labeling[25] (Fig. 6C in the reference) were performed according to their provided protocols (Supplementary Fig. 4d, e). The materials that we used were D-sorbitol (nacalai tesque #32021-95), glycerol (nacalai tesque #17018-25), methyl-β-cyclodextrin (TCI #M1356), γ-cyclodextrin (FUJIFILM Wako #037-10643), N-acetyl-L-hydroxyproline (Santa Cruz #sc-237135), and dimethylsulfoxide (nacalai tesque #13445-45) for AbScale, Tween-20 (nacalai tesque #35624-15), dimethylsulfoxide (nacalai tesque #13445-45), donkey serum (Sigma-Aldrich #D9663), glycine (TCI #G0317), heparin (Sigma-Aldrich #H3393), methanol (nacalai tesque #21915-93), hydrogen peroxide 30% (nacalai tesque #18411-25) and dichloromethane (Sigma-Aldrich #270997) for iDISCO+, and SDS (nacalai tesque #31606-75) for SWITCH. We used the antibodies as follows: A488-conjugated mouse anti-NeuN (5 μg/mL) in 1.6 mL of AbScale solution (0.33 M urea and 0.5% Triton X-100 in PBS) for the original AbScale protocol, mouse anti-NeuN (10 μg/mL) in 1.6 mL of primary staining buffer (5% DMSO, 3% donkey serum, 0.2%(v/v) Tween-20, 10 μg/mL of heparin in PBS) followed by anti-mouse secondary IgG (A488) (10 μg/mL) in 1.6 mL of secondary staining buffer (3% donkey serum, 0.2%(v/v) Tween-20, 10 μg/mL of heparin in PBS) for the original iDISCO+ protocol, and a complex of mouse monoclonal anti-NeuN IgG (10 μg/mL) and anti-mouse secondary Fab (A488) (1:1 weight ratio) in 0.5 mM SDS in PBS (SWITCH-Off buffer) or HEPES-TSC with 2.5% Quadrol for the SWITCH-mediated antibody labeling or CUBIC-HV, respectively. PBST was used as the SWITCH-On buffer.

**Klüver–Barrera (K–B) staining**. The formalin-fixed, paraffin-embedded human cerebellar tissue was sectioned to 4 μm thickness. K–B staining was performed with a standard protocol[58]. Images were acquired with a CCD camera (DP73, Olympus) attached to a microscope (BX53; UPlanSApo x40, NA: 0.95, Olympus).

**Measurement of EGFP signals**. To exclude the possibility of quenching during the staining procedure, we measured the EGFP fluorescent signals in the indicated reagents (Supplementary Fig. 3e). The method was performed according to the procedures from our previous study[40]. In brief, recombinant EGFP (50 μg/mL) was prepared in each reagent in 96-well plates and incubated for 24 h at 37 °C with gentle shaking. The 509 nm emission by 475 nm excitation was measured with a plate reader (EnSpire, PerkinElmer) ($n$ = 3 for each condition). The values were calculated as the means ± SD of the ratio of fluorescent signals before and after incubation (24 h/0 h) from three independent experiments.

**Imaging**. The 2D-sectioned samples were imaged with upright fluorescence microscopy (BX51, Olympus) with an objective lens (UPlanSApo 4×, N.A. = 0.16 and 20×, N.A. = 0.75), appropriate pairs of the filter and dichromatic mirror, and an sCMOS camera (ORCA-Flash4.0, Hamamatsu Photonics). The 16-bit image covering the preparation was acquired with a motorized $x$–$y$ stage (PRIOR) and tiled with the operational software (cellSens Dimension 1.18, Olympus).

The PI-stained and optically cleared marmoset body were imaged with a custom-built macrozoom light-sheet microscope (MVX10-LS, developed by Olympus) as described in our previous studies[19,36,40]. We used a 0.63× objective

lens (MV PLAPO 0.63×, N.A. = 0.15, W.D. = 87 mm, Olympus) with 1× optical zoom, a 532 nm laser, a bandpass filter (640/80 nm, φ 32 mm), and the tiling function of the operational software. During image acquisition, the optically cleared sample was immersed in an RI-matched oil mixture composed of silicone oil (HIVAC-F4, RI = 1.555, Shin-Etsu Chemical) and mineral oil (RI = 1.467, Sigma-Aldrich #M8410), and the final RI was adjusted to that of the cleared sample (~1.52). Sixteen-bit images were collected by scanning the sample in the z-direction with a 100 μm step size.

All of the other macroscopic images of the 3D stained mouse, marmoset, and human brain specimens were imaged with our GEMINI system, another custom-built light-sheet microscope combined with left and right light-sheet illumination units (developed by Olympus) and two macrozoom microscopes (MVX10, Olympus) placed at the front and back sides of the sample. Each microscope was equipped with the same objective lens as above (MV PLAPO 0.63×), optical zoom (0.63× to 4× range was used), a manual emission filter chamber, a tube lens (MVX-TV1XC, Olympus), and an sCMOS camera (Zyla 5.5, Andor). A series of φ 32 mm single bandpass filters (520/44, 585/40, 628/32, 641/75, and 708/75 nm, Semrock) were used to acquire signals from green to the near-infrared range. The excitation light source, provided by a fiber-coupled diode or DPSS lasers (Omicron, SOLE-6 equipped with λ = 488, 532, 594 and 642 nm), was connected to the light-sheet illumination units through a collimator and a beam-reflection mirror to switch between the left and right light paths. The sheet illumination was generated by the light-sheet illumination unit with a cylindrical lens. The thickness of the sheet illumination could be adjusted with a mechanical slit in the range of ~5–10 μm. Gel-embedded samples were placed onto a custom-made sample holder. The holder was connected to a motorized $x$–$y$–$z$ stage ($x$ and $y$ stages, Thorlabs, MTS50/M-Z8E; $z$ stage, Physik Instrumente, M-112.1DG) and immersed in an RI-matched oil mixture composed of HIVAC-F4 and mineral oil (RI: ~1.51) in a custom-made sample chamber with illumination and imaging windows. All electronic devices were controlled by custom-written LabVIEW software (National Instruments). Sixteen-bit images were collected by scanning the sample in the z-direction with a step size of 3–16 μm depending on the optical zoom of MVX10. A sheet illumination thickness of 5 or 10 μm was selected according to the step size. We collected the z-stack of the whole mouse brain and human cerebellar specimen from a single direction (front-to-back or back-to-front) because the clearing efficiency was sufficient for covering all of the sample areas without a visible decay of image quality. For the whole marmoset hemisphere, we used the two microscopes (placed on either side of the sample) to complement the image quality of deeper regions. We also applied an image acquisition sequence to achieve a homogeneous light-sheet thickness throughout the field of view in each $x$–$y$ image[59]. Thus, we collected several z-stacks by shifting the focus of the sheet illumination so that the confocal parameter of the beam waist covered the whole sample area of each $x$–$y$ image after tiling.

We determined the resolution of the images collected by GEMINI as follows. The typical voxel size of the data was 8.3 or 6.5 μm in the $x$–$y$ dimensions and 9 or 7 μm in the z dimension (using a 0.63× objective with 1.25× or 1.6× optical zoom, respectively). Here, the voxel size was calculated by the pixel size of the sCMOS sensor (6.5 μm²/pixel) and the step size of the z-axis (9 or 7 μm). According to the Nyquist–Shannon sampling theorem, the resolving power based on the voxel size can be determined as 16.6 or 13 μm for the $x$–$y$ dimensions and 18 or 14 μm for the z dimensions. Since these values were larger than the optical resolution of the microscope ($x$–$y$: ~6.6 or 4.2 μm at λ = 550 nm, provided by Olympus, z: ~10 μm at λ = 550 nm, based on the effective N.A. for light-sheet illumination ~0.03), the voxel size of the images accounts for the resolution of the data. Since CUBIC-R+ expands the sample almost 1.5-fold, the corresponding voxel resolution was ~11.1 or 8.7 μm for the $x$–$y$ dimension and 12 or 9.3 μm for the z dimension before clearing. The resolution range is similar to the range of cell body and nucleus sizes and is thus sufficient for resolving each nucleus in the images (Fig. 3j) as well as for computationally detecting sparsely labeled cell bodies/nuclei (Supplementary Figs. 9 and 10).

For highly magnified imaging of 3D synaptophysin staining samples (Supplementary Fig. 6n), an inverted confocal microscope (IX83, FV3000 system, Olympus) equipped with a 60× objective lens (UPLSAPO60XO, N.A. = 1.35, W.D. = 0.15 mm, Olympus) and a 594 nm laser were used. A single 16-bit $x$–$y$ image was acquired with 2× digital zoom (FOV: 106.1 × 106.1 μm², 4096 × 4096 pixels, 25.895 nm/pixel).

**Image processing**. The whole mouse brain images acquired by the GEMINI system were processed as follows. First, six focus-shifted images (three positions × left and right illuminations) at each z position were tiled to produce a single image. The x position of the tiling edge was determined by comparing the signal contrast (corresponding to the degree of sheet focusing) in the neighboring focus-shifted $x$–$y$ images at a representative z position. In most cases, we automatically calculated the tiling x position by using the nuclear staining images at the middle z position of the sample with a custom-written ImageJ macro. In case the algorithm did not properly work due to insufficient contrast in the images, we manually determined the tiling x position. Before tiling, the mean intensity among the images was equalized. After tiling, the resulting 16-bit images were converted to 8-bit, and blank regions were trimmed to reduce the file size. The noise signals in the images, attributed to particles in the embedding gel or aggregations of antibody, were

occasionally removed by filters as follows: (1) noise pixels outside the brain area were selected according to an intensity threshold, and the "Remove outlier" function was applied, (2) noise pixels were selected according to an intensity threshold higher than that of the staining signals, and the "Remove outlier" function was applied or the intensity was replaced with zero, or (3) noise pixels were selected with a combination of an intensity threshold and manual selection, and the "Clear function" was applied (Supplementary Data 2). Finally, the $x$–$y$ positions among the different channels were manually adjusted if needed.

To estimate light-sheet absorption in the tissue (Supplementary Fig. 4g), five consecutive images were selected from the middle of the z-stack of NeuN-stained whole mouse brain data. These images were projected with the "Maximum intensity" function, and the "Subtract Background" function (radius = 70) in Fiji/ ImageJ was then applied to separate the signal from the background. The light-sheet absorption was evaluated by comparing the intensity distribution of the lateral and medial parts of the background image. The collected intensity profiles were plotted with R (version 3.6.1).

To enhance the c-Fos staining signals (Fig. 8c; Supplementary Fig. 10b), we set an intensity threshold for deleting the background signals and then applied a 2D maximum filter (radius = 1), 2D Gaussian blur (sigma = 1), and a convolution filter with the kernel defined below:

$$k = \begin{pmatrix} -1 & -1 & -1 & -1 & -1 \\ -1 & -1 & 3 & -1 & -1 \\ -1 & 3 & 7 & 3 & -1 \\ -1 & -1 & 3 & -1 & -1 \\ -1 & -1 & -1 & -1 & -1 \end{pmatrix}.$$

For the human cerebellar specimen (Supplementary Fig. 11), four focus-shifted images (two positions × left and right illumination) at each $z$ position were collected. The images at each illumination were tiled, and the resulting left and right tiled images were then merged with the "Max intensity" function. The noise removal filters described above and adjustment of the $x$–$y$ position among the channels were also applied.

For the marmoset brain hemisphere (Fig. 9 and Supplementary Fig. 11), six focus-shifted images (three positions × left and right illumination) at each $z$ position were collected. The images of each illumination were tiled, and the resulting left and right images were then merged with the "Max intensity" function. For the whole hemisphere image (16.4 × 16.4 × 16 μm³ voxel resolution), the z-stacks were collected from two imaging directions (lateral-to-medial and medial-to-lateral) to complement the image quality of the deeper regions. Therefore, the left–right merged images at the same $z$ position acquired from the opposite directions were registered with the Descriptor-based registration (2D/3D) plugin (https://github.com/fiji/Descriptor_based_registration/) and the Transform J plugin[60] (https://github.com/imagescience/TransformJ/). Then, the registered images were merged with the "Max intensity" function. The noise removal filters described above and adjustment of the $x$–$y$ positions were also applied. In particular, we applied the "Subtract Background" function (radius = 10) to the α-SMA channel images due to the low SBR. We also stained the other hemisphere with the same antibody and confirmed that the low SBR result was specific for the single experiment rather than the staining protocol itself.

To obtain magnified images of these samples, we used a thinner (~5 μm) light sheet with a narrower confocal parameter. Thus, four to five images with different focused positions of either the left or right illumination direction were used for tiling. The tiled image was similarly processed as above.

For the LSFM images, the imaging and processing conditions are summarized in Supplementary Data 2. The z-stack data, acquired by LSFM and processed as above, were reconstituted and visualized as a pseudocolored image with Imaris software (version 8.4, Bitplane). The intensity and gamma values were manually adjusted to prepare figure panels. Computational processing of other acquired images was performed by using Fiji/ImageJ. The intensity is adjusted with the brightness and contrast functions. Occasionally, some images were cropped, and their orientations were adjusted to prepare figure panels. The reconstituted $x$–$z$ or $y$–$z$ images in Fig. 3j and Supplementary Figs. 5a, 6g, and 9c were prepared with the reslice function.

**Fusion of 10 different whole-brain images.** Our virtual multiplex registration pipeline[40] (https://github.com/DSP-sleep/Landscape_pipeline/wiki) was used to combine ten different whole-brain images. The nucleus channel of each whole-brain dataset stained with a nuclear stain (SYTOX-G or BOBO-1) and an antibody (anti-calbindin D28K, PV, Sst, Th, ChAT, Dbh, Tph2, Copeptin, or phospho-Nf) was registered and aligned to the nucleus channel of an anti-NeuN-stained whole-brain dataset with the symmetric normalization (SyN) algorithm implemented in the ANTs software[61]. The obtained transformation matrix was applied to the antibody-stained channel without losing resolution. The resulting image stacks were reconstituted with Imaris.

**Computational cell detection.** Labeled cells were computationally detected by using Fiji/ImageJ as follows (Supplementary Figs. 9 and 10). For RV-labeled data, (0) an image stack for each channel cropped around the injection site was prepared; (1) the original stack was divided by duplicated and Gaussian-blurred stacks

(sigma = 1) to remove the background (this step was optional and used only for the RV-GFP and Sst-Cy3 images); (2) the "Subtract Background" function (rolling = 1) was applied to the stack; (3) the stack was expanded three times in the $x$–$y$ directions to increase detection accuracy; (4) a Gaussian blur filter (sigma = 2) was applied; (5) a Laplacian filter (sigma = 2) from the Feature J plugin (included in The ImageScience library) was applied. The resulting stack was converted from 32-bit to 16-bit; (6) a minimum filter (radius = 1.5) was applied (this step was optional and used only for RV-GFP and AAV-TVA-mCherry images); (7) the "Set Threshold" function with a manually adjusted value was applied to eliminate background signals. A different value was determined for each channel; (8) the "2D Find Maxima" function was applied with a manually adjusted noise tolerance value. A stack with single pixels of maxima positions was generated; (9) the Maximum filter was applied to enlarge the maxima points from one pixel to five pixels; (10) the 3D centroids of the maxima objects were calculated by the "3D Object Counter" function. The size filters were as follows: 6–76 pixels for RV-GFP and AAV-TVA-mCherry and 6–36 pixels for Sst. The results were output as a stack with centroid spots and a CSV file of the result table. The centroid stack was resized to the original $x$–$y$ dimensions; and (11) a Gaussian blur filter was finally applied to the stack to adjust the spot size to the labeled cell size. The calculations for 11.3 MB of cropped data were completed within a few minutes by using a 2018 15-inch MacBook Pro (macOS High Sierra version 10.13.6, 2.9 GHz Intel Core i9 processor, and 32 GB 2400 MHz DDR4 memory).

For the c-Fos-labeled whole-brain data, we applied the "3D Find Maxima" function rather than the 3D centroid calculation for 2D Maxima objects adopted in the ClearMap algorithm[17]. The same parameter values were used for the two datasets (MK-801 + or −) of the c-Fos-labeled brains. 0) The original stack covering the whole brain was cropped to prepare 100 partitioned stacks with 10-pixel $x$–$y$ margins; (1) the "Subtract Background" function (rolling = 1) was applied to the single stack; (2) the stack was expanded three times in the three dimensions ($x$–$y$–$z$); (3) a 3D Minimum filter (radius $x = y = z = 1$) was applied; (4) a 3D Gaussian blur filter (sigma $x = y = z = 1$) was applied; (5) the "Set Threshold" function with a manually determined value was applied to eliminate background signals; (6) the "3D Find Maxima"[62] function was applied with a radius size of $x = y = z = 2.5$ and a noise tolerance of 1. A stack with single pixels of maxima positions and a CSV file of the result table were generated. Every 120 slices, 5 marginal slices were separately calculated and integrated into the single stack generated above; (7) after calculating all 100 stacks, they were resized to the original $x$–$y$–$z$ dimensions and then tiled after removing the 10-pixel $x$–$y$ margins; and (8) the 3D Maximum filter and 3D Gaussian blur filter were finally applied to the stack to adjust the spot size to the labeled nucleus size. It took ~20 h to perform the calculations for ~8 GB of whole-brain data by using a workstation PC (64-bit Windows 10 Pro for Workstations, Intel(R) Xeon(R) E5-2687W v4 @ 3.00 GHz, two processors, 192 GB RAM, and 4× SSD RAID0 storage).

We calculated the sensitivity (number of cells correctly detected by the algorithm/number of cells in the ground truth data) and the positive predictive value (number of cells correctly detected by the algorithm/total number of detected cells by the algorithm). The ground truth data were prepared by manual segmentation of the labeled cells with ITK-SNAP[63]. The cells correctly detected by the algorithm were determined by comparing the coordinates of the cells (centroid/ 3D Maxima) with those of the ground truth. Here, we selected regions where computational cell detection was likely difficult due to the high cell density: for the RV- and Sst-labeled brains, we used the region already shown in the panels (Supplementary Fig. 9b-c) to detect the double- (RV-GFP+, Sst+) or triple- (AAV-mCherry+, RV-GFP+, Sst+) labeled cells. For the Gad2-Cre strain, RV labeling depicts the local network of GABAergic neurons around the injection site. In contrast, such GABAergic input in other cortical regions (e.g., the somatosensory cortex) is rare [Fig. 7e and the mouse connectivity map of the Allen brain atlas, #167441329 (AAV-labeled projection S1 barrel of Gad2-IRES-Cre strain), http:// connectivity.brain-map.org]. Therefore, it seems reasonable to focus on the region rather than covering a larger area. Similarly, we used a cortical region from the MK-801-treated mouse brain to calculate the sensitivity and PPV of c-Fos-positive cell detection (Supplementary Fig. 10c). The sensitivity and positive predictive value (PPV), respectively, of the algorithm are as follows: 114/128 = 89.1% and 114/115 = 99.1% for GFP+ cells; 54/73 = 74.0% and 54/55 = 98.2% for mCherry+ cells; 110/169 = 65.1% and 110/115 = 95.7% for Sst+ cells; and 382/491 = 77.8% and 381/381 = 100% for c-Fos+ cells.

**Statistics and reproducibility.** For Supplementary Fig. 3e, one-way ANOVA was applied to the data after the confirmation of nonnormality and unequal variance with the Kolmogorov–Smirnov test and Bartlett's test, respectively. In this study, $P < 0.05$ was considered significant. R (version 3.4.4) was used for the statistical analyses.

The quantification of proteins and lipids in the brain and the swelling-shrinkage assays of the gels and delipidated tissues were repeated with at least two independent samples (Fig. 1 and Supplementary Fig. 1). The SAXS analyses were performed with independent samples and detectors (SPring-8 and PF of KEK) (Fig. 1). Most of the staining and imaging experiments shown in Figs. 2–9 and Supplementary Figs. 2–11, except for the marmoset hemisphere staining, were repeated with at least two independent samples in the same or comparable condition with slight modification (e.g., staining period). All the results were

reliably reproduced. Since we obtained only one marmoset brain (two hemispheres) and the other hemisphere was used to determine the 3D staining condition with anti-α-SMA antibody, we could not carry out another replication experiment for GFAP staining.

**Reporting summary**. Further information on research design is available in the Nature Research Reporting Summary linked to this article.

## Data availability

The data that support the findings of this study are available from the corresponding author upon reasonable request.

## Code availability

The codes for the custom-written ImageJ macro and the virtual multiplex registration pipeline[40] were deposited in our GitHub repositories (https://github.com/DSP-sleep/CUBIC-HistVIsion.git, https://github.com/DSP-sleep/Landscape_pipeline/wiki). The custom-made software for SAXS measurement data is available from the corresponding authors upon reasonable request.

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

## Acknowledgements

We thank all of the laboratory members at The University of Tokyo and the RIKEN Center for Biosystems Dynamics Research (BDR), in particular, Y. Liu, R. Tanaka, Y. Saito, Y. Ueki, S. Yada and R. Ohno for their support in sample preparation and experiments; T.C. Murakami and A. Kawasaki for help in clearing the infant marmoset body; H. Fujishima and K. Matsumoto for help in introducing suitable antibody products; T. Mano and M. Kuroda for improving the GEMINI system and for their helpful suggestions for image analysis, S. Shi for helping with the statistical analysis, and T. Miyawaki and K. Wilkins for assisting with the preparation of the manuscript. We also thank Dr R. Yoshida for helpful suggestions regarding gel materials, Dr N. Mizushima for support in the measurement of EGFP signals, Olympus Corporation, ISAX Corporation and CREDEN TECHNOLOGIES SDN BHD for helping with the GEMINI system installation, and Bitplane for instruction in operating the Imaris software. This work was supported by PRESTO from JST (JPMJPR15F4 to E.A.S.); Grants-in-Aid for Young Scientists (A) (JSPS KAKENHI grant 15H05650 to E.A.S.); Grants-in-Aid for Scientific Research (B) (JSPS KAKENHI grant 19H03413 to E.A.S. and 18H02105 to K. Tainaka); Grants-in-Aid for Scientific Research on Innovative Areas (JSPS KAKENHI grant 17H06328 to E.A.S., 17H05688 and 18H04543 to K. Tainaka); Grants-in-Aid for Challenging Research (Exploratory) (JSPS KAKENHI grant 18K19419 to E.A.S. and 18K19373 to K. Tainaka); Grants-in-Aid from the Takeda Science Foundation (H.R.U., E.A. S., and K.Tainaka); Grants-in-Aid from the Japan Foundation for Applied Enzymology (E.A. S.); AMED-CREST (AMED/MEXT to H.R.U.) and CREST (JST/MEXT to H.R.U.); Brain/MINDS (AMED/MEXT to H.R.U. and T.I.); the Basic Science and Platform Technology Program for Innovative Biological Medicine (AMED/MEXT to H.R.U.); the Science and Technology Platform Program for Advanced Biological Medicine (AMED/MEXT to H.R. U.); Grants-in-Aid for Scientific Research (S) (JSPS KAKENHI grant 18H05270 to H.R.U.); the Collaborative Research Project of the Brain Research Institute, Niigata University (to H. R.U.); Grants-in-Aid from the Human Frontier Science Program (H.R.U.); Grants-in-Aid for Scientific Research (A) (JSPS KAKENHI grant 16H02277 to M.S.); Grants-in-Aid for Scientific Research (JSPS KAKENHI grant 19J13188, to H. Ono); and the ERATO Touhara Chemosensory Signal Project from JST, Japan (JPMJER1202 to K. Touhara). The *App*[NL-G-F] knock-in mouse strain (RBRC06344) was provided by RIKEN BRC through the National BioResource Project of the MEXT/AMED, Japan. The SAXS experiments were performed at the Frontier Soft Matter Beamline (FSBL; BL03XU), SPring-8, Hyogo, Japan, with the assistance of Dr Hiroyasu Masunaga and Dr Atsushi Izumi (Proposal No. 2015B7260, 2016A7210) under the approval of the Photon Factory Program Advisory Committee (Proposal No. 2015G134).

## Author contributions

E.A.S. and H.R.U. conceived and designed the study. E.A.S. and A. Kuno. initialized the project. E.A.S and C.S. performed most of the experiments and the analysis. A. Kuno, X.L., K.N., and M.S. performed the SAXS experiments. K.Tainaka measured the components of the tissue. K.L.O. and Y.Saeki conducted the simulation of the reaction-diffusion formula in the gel. K.M. and K. Touhara provided the Gad2-Cre KI mouse brains labeled with the RV system. K.I. and T.I. provided the adult marmoset brains. C.Y. and H. Onoe provided the infant marmoset body. H.K. and A. Kakita provided the human postmortem cerebellum specimens and several antibodies for staining tests and performed the K–B staining. M.I., T.U., and M.F. provided the human postmortem brain specimens for determining clearing conditions. Y. Shimizu provided the recombinant EGFP protein. T.S. and T.C.S. provided the *App*[NL-G-F] knock-in mice, and H. Ono stained the brains. E.A.S. and H.R.U. wrote the manuscript. All authors discussed the results and commented on the manuscript text.

## Competing interests

RIKEN has filed a patent regarding CUBIC reagents, in which E. A. S., K. Tainaka, and H. R. U. are co-inventors. CUBICStars CO., LTD. has filed a patent regarding CUBIC-HV reagents, in which E. A. S., and H. R. U. are co-inventors. Part of this study was performed in collaboration with Olympus Corporation and Medical & Biological Laboratories Co., Ltd. (MBL). The remaining authors declare no competing interests.

## Additional information

Etsuo A. Susaki [1,2✉], Chika Shimizu[2], Akihiro Kuno [1,3], Kazuki Tainaka[4], Xiang Li[5], Kengo Nishi[5], Ken Morishima [5], Hiroaki Ono[1], Koji L. Ode[1,2], Yuki Saeki[6], Kazunari Miyamichi[7,8], Kaoru Isa[9,10], Chihiro Yokoyama[11], Hiroki Kitaura[12], Masako Ikemura[13], Tetsuo Ushiku[13], Yoshihiro Shimizu [14], Takashi Saito[15,16], Takaomi C. Saido[15], Masashi Fukayama[13], Hirotaka Onoe[17], Kazushige Touhara [7,8,18], Tadashi Isa[9,10], Akiyoshi Kakita[12], Mitsuhiro Shibayama[5] & Hiroki R. Ueda [1,2✉]

[1]Department of Systems Pharmacology, Graduate School of Medicine, The University of Tokyo, 7-3-1 Hongo, Bunkyo-ku, Tokyo 113-0033, Japan. [2]Laboratory for Synthetic Biology, RIKEN Center for Biosystems Dynamics Research, 1-3 YamadaokaSuitaOsaka 565-5241, Japan. [3]Department of Anatomy and Embryology, Faculty of Medicine, University of Tsukuba, 1-1-1 Tennodai, Tsukuba 305-8575, Japan. [4]Department of System Pathology for Neurological Disorders, Brain Research Institute, Niigata University, 1-757 AsahimachidoriChuo-kuNiigata 951-8585, Japan. [5]Neutron Science Laboratory, The Institute for Solid State Physics, The University of Tokyo, 5-1-5 KashiwanohaKashiwaChiba 277–8581, Japan.

[6]Faculty of Medicine, The University of Tokyo, 7-3-1 HongoBunkyo-kuTokyo 113-0033, Japan. [7]Department of Applied Biological Chemistry, Graduate School of Agricultural and Life Sciences, The University of Tokyo, 1-1-1 YayoiBunkyo-kuTokyo 113-8657, Japan. [8]ERATO Touhara Chemosensory Signal Project, Japan Science and Technology Agency, The University of Tokyo, 1-1-1 YayoiBunkyo-kuTokyo 113-8657, Japan. [9]Department of Neuroscience, Graduate School of Medicine and Faculty of Medicine, Kyoto University, Yoshida-konoe-cho, Sakyo-kuKyoto 606-8501, Japan. [10]Institute for the Advanced Study of Human Biology (WPI-ASHBi), Kyoto University, Kyoto 606-8501, Japan. [11]Laboratory for Brain Connectomics Imaging, RIKEN Center for Biosystems Dynamics Research, 6-7-3 Minatojima-minamimachi, Chuo-ku, KobeHyogo 650-0047, Japan. [12]Department of Pathology, Brain Research Institute, Niigata University, 1-757 AsahimachidoriChuo-kuNiigata 951-8585, Japan. [13]Department of Pathology, Graduate School of Medicine, The University of Tokyo, 7-3-1 HongoBunkyo-kuTokyo 113-0033, Japan. [14]Laboratory for Cell-Free Protein Synthesis, RIKEN Center for Biosystems Dynamics Research, 6-2-3, Furuedai, SuitaOsaka 565-0874, Japan. [15]Laboratory for Proteolytic Neuroscience, RIKEN Center for Brain Science, 2-1 HirosawaWakoSaitama 351-0198, Japan. [16]Department of Neurocognitive Science, Institute of Brain Science, Nagoya City University Graduate School of Medical Science, 1 Kawasumi, Mizuho-cho, Mizuho-ku, Nagoya, Aichi 467-8601, Japan. [17]Human Brain Research Center, Graduate School of Medicine, Kyoto University, 54 Shogoin-kawahara-choSakyo-kuKyoto 606-8507, Japan. [18]International Research Center for Neurointelligence (WPI-IRCN), UTIAS, The University of Tokyo, Tokyo 113-0033, Japan. ✉email: suishess-kyu@umin.ac.jp; uedah-tky@umin.ac.jp

