## [Peer Review File · Nature Communications]

Reviewers' comments:

Reviewer #1 (Remarks to the Author):

Dear Editor,

I carefully read the manuscript "Versatile whole-organ/body staining and imaging based on electrolyte-gel properties of biological tissues" by Susaki and co-workers.

The Authors convincingly demonstrate that brain tissue can be modeled as an electrolyte gel, and use this finding to optimize staining protocols with both histological labels and antibodies. The applicability of the optimal protocol on brain samples from different species (mouse, marmoset human) clearly demonstrates the potential of this approach.

This work is definitely a significant contribution to the field of whole-mount staining methods, as it not only reports useful protocols but lays the foundations for a more rational design of labeling strategies. In this view, given the large general interest raised by clearing and labeling of macroscopic specimens, I found this manuscript well suited for the broad audience of Nature Communications.

However, several issues need to be properly addressed before final acceptance:

1. The experiments shown support modeling brain tissue as an electrolyte gel. However, this may not hold for other tissues (muscles, liver, kidney, etc.). The Authors must clearly disclose that the scope of the work is limited to brain tissue, and avoid referring to general "biological tissue" (as in current title, which I suggest to modify to "brain tissue" or similar). Furthermore, I suggest to add some discussion about extending the approach presented in the present paper to other types of tissue. This would underline the potential of the modeling paradigm presented.
2. The analysis performed in human cerebellum need further support. Indeed, I feel that a validation of effective antibody penetration – as presented in mouse (e.g. Suppl. Figs. 3 and 4) – is lacking. Without such validation, one could argue that the increased GFAP staining in the molecular layer is an artefact due to poor penetration of the antibody. A similar validation would be important also in the marmoset.
3. I think that the overall impact of the work would be greater if the Authors discuss possible application of their modeling approach also to the various implementations of expansion microscopy.
4. The conclusion that immunolabeled cells could be computationally detected should be supported also by some quantification of the precision and sensitivity (or number of false positives/negatives) of the detection algorithm. Data shown in Suppl. Fig. 6c looks very nice, but some numbers over a larger area would pose this result on a more solid basis.
5. In the Discussion (page 38, line 4). The Authors should be more precise in describing which aspects of SWITCH did not work as expected in their hands.
6. Given the resolution of the imaging, it is hard to say that synaptic structures can be visualized (Page 26, line 1-2). I would personally remove that specific example.
7. I think that at the end of page 28, the Authors should refer also to the paper by Ye et al. (Cell 2016) as an example of full 3D activation mapping.
8. Did the Authors test any other antibody on human or marmoset? Is there any specific reason in the choice of the antibodies used (rather than e.g. NeuN)?
9. The gel / solution drawings in Fig. 1G are not very clear to interpret. I suggest the Authors to better distinguish free ions in the solution from ions attached to the gel – maybe using different colors.
10. Scale bars in figures 2b-e or 3a-e. I guess that the actual size of gel or tissue will change according to different experimental conditions. Thus, each image would require its own scale bar.
11. Typo in fig. 3e: "appropreate" should be "appropriate"
12. Methods, page 63, line 11-14. The patient from which the cerebellum was extracted by surgery is the same that suffered from amyotrophic lateral sclerosis? This is not clear in the current text.

Reviewer #2 (Remarks to the Author):

In "Versatile whole-organ/body staining and imaging based on electrolyte-gel properties of biological tissues", Susaki et al. present work on developing an improved pipeline for staining and immunostaining cleared tissues using variations on the CUBIC methodology. The manuscript presents several principal findings: first, they show some relatively fundamental results that tissues (particularly brain) behave as a hydrogel using some classical physical chemistry assays (i.e., SAXS); second, they develop an assay for optimization of stain uniformity with small molecule stains of nucleic acids and antibody stains, and they use this assay together with their insights into tissues acting as a hydrogel to optimize conditions that favor uniform staining; third they demonstrate optimized labeling of a range of specimens. The high quality of the 3D images and demonstrations give an impression of a robust and useful method for the community. They are generally clear and convincing. Overall, I think the work is well-suited for Nature Communications and would support its publication subject to the suggestions, below.

1. In their electrolyte gel model, the positively charged nuclei dye, PI has strong interactions with the negatively charged gel and a high salt concentration improve the uniformity of labeling. The authors also demonstrate that high salt improves the uniformity of lipophilic dyes. Could the authors comment on the staining conditions likely to favor uniform labeling with a negatively charged dye? This would be interesting considering the authors' statement that "most of the cationic amino residues had been masked by PFA fixation and the isoelectric points of CUBIC-delipidated brain was around pH 5, more abundant ionized residues in the polymer were thought to be the anionic carboxyl group."

2. I appreciate the authors' comments in the discussion about some limitations of this approach for the study of brains. However, I wonder whether other organs do not work as well as brains. For instance, in Fig. 2h, the nuclei stain from other organs is not as uniform as in brain. Could the authors comment on this?

3. Do the authors have any issues with substantial absorption of light through thick tissues? This is sometimes referred to as the "inner filter effect" and could potentially be quite strong for thick tissues.

4. Aspects of the presentation could be improved.

a. I feel the article was a bit longer than necessary to communicate the central points, but provide this feedback merely as a general suggestion.

b. On P13L8 (page 13, line 8) the authors mention four-fold shrinkage of tissue, and they mean tissue volume, right? This should be made clear that it is not linear four-fold shrinkage which, on three axes, would be 64-fold shrinkage of tissue.

c. While this reviewer is sensitive that authors may not be native English speakers, there are nonetheless many typos or odd phrases. The language could use a fair amount of polishing ("Biological researches" on P6L5, "Does not penetration well" on P7L14, "Observed in the HEPES with" on P17L5, "Custom build" instead of custom-built on P18L3, and many others... I did not create a full list).

d. The results of this paper seem somewhat to contradict the results of the SWITCH approach from Kwanghun Chung's group at MIT. The authors here state they couldn't make SWITCH work. Perhaps there are aspects of the SWITCH protocol that haven't been followed or that may not have originally been thoroughly described, but we are not in a position to judge from the brief statement. Overall the treatment of SWITCH feels a bit dismissive and critical without concretely explaining why. I think that a more careful discussion on this topic would be helpful and of great interest to practitioners. For instance, the following statement (P38L2) was vague in referring to

SWITCH. "However, there were still potential concerns regarding on the two-step implementation and the choice of SDS without evaluation for broader antibodies in the study." What were the potential concerns to which the authors are now referring? In the SWITCH paper, they studied ~100 antibodies against ~40 different proteins, almost all of which were reported to work well with the SWITCH procedure. That is an equal or larger number of antibodies tested than in the manuscript under review, isn't it?

e. The quality of the movies was low. The authors should upload higher resolution movies that can properly show off their high-quality data. Hopefully there are no journal restrictions (such as low file size limits or poor default suggestions in the file preparation instructions) that are causing this.

Reviewer #3 (Remarks to the Author):

The present work presents a highly powerful and convincing 3D staining and imaging pipeline based on their previously established CUBIC approach. The results of this workflow, which the authors denote as CUBIC-HV is illustrated with many compelling experiments images.

The concept of describing the physicochemical properties of biological tissue as an electrolyte gel of cross-linked polypeptides is innovative, and the experiments leading to this conclusion are convincing. This is a particular strength of the current work that it rationalizes the otherwise often alchemistic art of 3D labeling. A clear advantage of the approach is that it can really quantify the soft material properties of tissue and then uses this knowledge to optimize labeling.

While I cannot judge all aspects of the work, in particular since I lack expertise in histo-chemistry and do not know exactly all lines of current state-of-the art, I am impressed by the breadth and depth of experiments and data collection, all aligned towards a well thought and research design of high intellectual merit. I can strongly recommend publication from a more general point of view of 3D histology. The approach may provide benchmarking results. At the same time, I would encourage the authors to avoid over-statements, and carefully check all statements made, mainly in view of (a) structure preserving, (b) necessary resolution for sub-cellular detail and cellular identification, (c) imaging of brain circuitry.

At several instances when reading the work, I had strong impressions of over-claims. For example, concerning cellular identification.

1.) The voxel sizes are quite large in many of the results, and resolution was not determined at all (for example by FSC). This is clearly a deficit of the MS. The rather modes resolution must affect the recall and precision of cellular identification. Please comment this critically, wherever applicable in the MS.

2.) Closely related to this point: p. 29, l.13 and following: why does this experimental result prove identification of whole brain circuit structure, neuronal connections cannot be visualize in this manner;

3.) p 42, l.17.18, Suppl. Fig.4: Please explain more explicitly to which end the data shown in Fig.4 proves preservation of subcellular structure and molecular integrity

Another example, where I am sceptical to which end neurons are well identified is the example of human cerebellum.

4.) Suppl- Fig.7 . Please explain at which level neurons (recall, precision, dice score ect.) neurons in the molecular, Purkinje cell layer, and granular layer can be identified.

5.) SAXS curves of non-delipidated and delipidated tissue. The tentative attribution of the peak at 0.08 μm to myelin should be substantiated by comparison to literature, for example Carboni et al, Biomedical Optical Express 2017, who report a first peak at about 0.09 and a second at 0.16. There may be interesting conclusions/explanations regarding the osmotic pressure and other variable which may affect the d-spacing of myelin here.

6.) The interpretation of the SAXS curves in terms of a fractal geometry should be rephrased more carefully, which a fractal structure can explain the power laws, it cannot be uniquely be inferred from the data. A model and full q-range fit of the data would be required to really exploit the information in this data.

7.) The diffusion-reaction model of the staining process: Please add information of time scale and time units (simulation steps). A homogeneous distribution is clearly the steady-state solution of the equation, so all non-homogeneous pattern should only be solution while the systems converges to a steady state. Time scales (even if only in 'simulation units') is critical here and need to be described.

While it is very well written, typos and grammar have be checked throughout the MS.

Responses to Reviewers' Concerns for Susaki et al., "Versatile whole-organ/body staining and imaging based on electrolyte-gel properties of biological tissues".

We were delighted to receive the overall very positive and enthusiastic response from reviewers. We thank all reviewers for their insightful and constructive comments, and especially for their appreciation of the fundamental importance of our study. We are confident that we have addressed all the reviewers' concerns by providing new critical and significant experimental data and incorporating valuable suggestions in the revised manuscript.

In particular, we acquired macroscopic swelling-shrinkage behaviors of the kidney, liver, and muscle, and collected 3D images of whole kidney staining. We have also obtained 3D staining results of eosin, a typical negatively charged dye. We added and elaborated on the results of SWITCH-mediated antibody labeling. To exclude the possibility of the inner filter effect of light-sheet illumination, we quantified the background fluorescence of whole-brain imaging. We also evaluated the sensitivity and positive predictive value of the computational cell detection in Supplementary Fig. 6. The revised manuscript also presents new images of GFAP antibody staining on human and marmoset cerebellar sections, Klüver-Barrera (KB) staining of the human cerebellum, and a new video of marmoset whole body staining and image data. We thoroughly checked the manuscript for correcting spelling and expression. We also somewhat shortened the manuscript by deleting redundant sentences.

We added these new data and corresponding descriptions to the manuscript and made other changes to address the specific concerns raised by the reviewers. Details of the changes are as follows. We believe that these changes have significantly strengthened our manuscript for publication.

Response to Reviewer #1:

I carefully read the manuscript “Versatile whole-organ/body staining and imaging based on electrolyte-gel properties of biological tissues” by Susaki and co-workers. The Authors convincingly demonstrate that brain tissue can be modeled as an electrolyte gel, and use this finding to optimize staining protocols with both histological labels and antibodies. The applicability of the optimal protocol on brain samples from different species (mouse, marmoset human) clearly demonstrates the potential of this approach.

This work is definitely a significant contribution to the field of whole-mount staining methods, as it not only reports useful protocols but lays the foundations for a more rational design of labeling strategies. In this view, given the large general interest raised by clearing and labeling of macroscopic specimens, I found this manuscript well suited for the broad audience of Nature Communications.

We are glad that this reviewer was enthusiastic about our study and described it as “definitely a significant contribution to the field of whole-mount staining methods” because it “not only reports useful protocols” but also “lays the foundations for a more rational design of labeling strategies.” We have appreciated the reviewer’s deep understandings of our study, the supportive suggestion for publication, and many constructive comments. We have been able to address all the points in full.

However, several issues need to be properly addressed before final acceptance:

1. The experiments shown support modeling brain tissue as an electrolyte gel. However, this may not hold for other tissues (muscles, liver, kidney, etc.). The Authors must clearly disclose that the scope of the work is limited to brain tissue, and avoid referring to general “biological tissue” (as in current title, which I suggest to modify to “brain tissue” or similar). Furthermore, I suggest to add some discussion about extending the approach presented in the present paper to other types of tissue. This would underline the potential of the modeling paradigm presented.

Comment #1: The reviewer raised a concern about whether the other tissues (eg., muscles, liver, kidney, etc.) could be modeled as electrolyte gels as brain tissue. The reviewer also suggested adding some discussion about extending the approach presented in the present paper to other types of tissue.

We appreciate the reviewer's comment. Given the mutuality of tissue composition (lipid, protein, nucleic acids and water, Fig. 1c), we assumed that other tissues could be modeled almost identically as the same electrolyte gel. To confirm the assumption, we observed the repeated and reversible ability of delipidated muscles, liver and kidney as was the case in the brain in Fig. 1a (new Supplementary Fig. 1h). In the assay, we tested their ability of 1) being swelled in the alkaline condition, 2) being shrunk in the high ionized condition, and 3) again being shrunk in the acetone condition, all of which were consistent with the electrolyte gel model (page 14, line 9-17). In addition, we successfully 3D-stained a delipidated kidney with SYTOX-G and anti-CREB (new Supplementary Fig. 4p) (page 26, line 14-17). These data further supported our assumption of the model expansion from the brain. According to the reviewer's suggestion, we also specified the use of the brain as a representative organ in each experiment (e.g., page 10, line 5-6) and added some discussion about extending our findings in the brain to other types of tissue (page 36, line 10-15; page 42, line 6-8).

2. The analysis performed in human cerebellum need further support. Indeed, I feel that a validation of effective antibody penetration – as presented in mouse (e.g. Suppl. Figs. 3 and 4) – is lacking. Without such validation, one could argue that the increased GFAP staining in the molecular layer is an artefact due to poor penetration of the antibody. A similar validation would be important also in the marmoset.

Comment #2: The reviewer also raised a concern about whether the increased GFAP staining in the molecular layer of the human cerebellum would be an artifact due to poor penetration of the antibody. The reviewer also commented that a similar validation would also be important in the marmoset.

We also appreciate the reviewer's comment. Following the reviewer's comment, we compared the GFAP staining results on the 2D section and 3D reconstituted section of the human cerebellum (new Supplementary Fig. 7b-f), as the validation in Supplementary Fig. 4. We observed a comparable staining pattern of GFAP signals in the molecular layer in both images, ruling out the possibility of poor antibody penetration. We also performed a similar comparison for the marmoset cerebellum and again confirmed relatively strong GFAP signals in the granular layer (new Supplementary Fig. 7h) as in the 3D reconstituted

image (Supplementary Fig. 7g). We addressed these results in the revised manuscript (page 34, line 6-9). To reflect the concern raised by the reviewer, we also mentioned such a potential issue on the dye/antibody penetration in primate tissues in Discussion of the revised manuscript (page 43, line 9-11).

3. I think that the overall impact of the work would be greater if the Authors discuss possible application of their modeling approach also to the various implementations of expansion microscopy.

Comment #3: The reviewer suggested that the overall impact of the work would be greater if we discuss the possible application of the tissue gel modeling approach also to the various implementations of expansion microscopy.

We thank for the reviewer's kind suggestion. According to the comment, we add discussion as follows:

1) the CUBIC-X protocol does not utilize artificial water-absorbing polymers for expansion but enhances the inherent gel properties of biological tissue (page 36, line 10-15), and 2) a similar protocol optimization might be possible based on chemical properties of acrylamide-based gel in CLARITY, PACT and ExM (page 41, line 13-16), both of which help increase the overall impact of this study.

4. The conclusion that immunolabeled cells could be computationally detected should be supported also by some quantification of the precision and sensitivity (or number of false positives/negatives) of the detection algorithm. Data shown in Suppl. Fig. 6c looks very nice, but some numbers over a larger area would pose this result on a more solid basis.

Comment #4: The reviewer suggested that the computational detection of immunolabeled cells should be supported by some quantification (e.g., precision, sensitivity or number of false positives/negatives) of the detection algorithm.

We thank the reviewer for raising the point. According to the reviewer's suggestion, we evaluated the sensitivity (number of correctly detected cells by algorithm/number of cells in the ground truth data) and the positive predictive value (number of correctly detected cells by the algorithm/total number of detected cells by the algorithm) of the algorithms. Here, we selected regions where the computational cell detection is plausibly difficult due to the high cell density: for the RV- and Sst-labeled brain, we used the region already shown in

the panels (Supplementary Fig. 6b-c) to detect the double- (RV-GFP+, Sst+) or triple- (AAV-mCherry+, RV-GFP+, Sst+) labeled cells. In the Gad2-Cre strain, RV labeling depicts the local network of GABAergic neurons around the injection site, while such GABAergic input out of other cortical regions (eg, somatosensory cortex) is apparently rare [Fig. 6e; mouse connectivity map of Allen brain atlas, #167441329 (AAV-labeled projection S1 barrel of Gad2-IRES-Cre strain), <http://connectivity.brain-map.org>]. Therefore, it seems reasonable to focus the region rather than covering over larger area. Similarly, for the sensitivity/PPV calculation of the cFos-positive cell detection, we used a cortical region of MK-801-treated mouse brain (indicated in the new Supplementary Fig. 6f).

These calculation results and documentations are added in the revised manuscript (in the legends of Supplementary Fig. 6) (page 97, line 11 – page 98, line 10). Although we think that the computational detection results here (PPV in particular) are sufficient for showing the immunolabeling quality by CUBIC-HV, we recognize the users' demand for a more sophisticated whole-organ cell detection and analysis method. We will address the point in our future manuscript.

5. In the Discussion (page 38, line 4). The Authors should be more precise in describing which aspects of SWITCH did not work as expected in their hands.

Comment #5: The reviewer suggested that we should more precisely describe which aspects of SWITCH did not work as expected in our hand.

We thank the reviewer's comment. To clarify the point that the reviewer pointed out, we added our data of anti-NeuN 3D staining by SWITCH-mediated antibody labeling [corresponding to Figure 6C in Murray et al. Cell (2015)] and CUBIC-HV (Supplementary Fig. 3j, k). We compared their two-step implementation (SWITCH-Off in 0.5 mM SDS and SWITCH-On in PBST) and our one-step implementation (simultaneous penetration and reaction in HEPES-TSC supplied by Quadrol). As shown in the new figure panels, we did not observe NeuN signals in the SWITCH-stained cerebellum, while we obtained the expected 3D staining result by CUBIC-HV on a similar timescale. In the revised manuscript, we described the results (page 23, line 17 - page 24, line 1) as well as more accurately discussed the potential issues of the tested SWITCH-mediated antibody labeling, such as the insufficient recovery of immune reaction in SWITCH-On condition (page 39, line 1-8). We also addressed the point in the reply to the comment #4-d by reviewer #2.

6. Given the resolution of the imaging, it is hard to say that synaptic structures can be visualized (Page 26, line 1-2). I would personally remove that specific example.

Comment #6: The reviewer raised a concern regarding the imaging resolution to visualize synaptic structures in Supplementary Fig. 4n-o.

We thank the reviewer's comment. As described in Methods (Imaging) and the legend of Supplementary Fig. 4o, we acquired the data with confocal microscopy equipped with a 60×/1.35 objective lens and 2× digital zoom (FOV : 106.1 × 106.1 μm, 4096 × 4096 pixels, 25.895 nm/pixel). The resolution was sufficient to observe the synaptophysin-labeled synaptic staining patterns. We further clarify the details in the main text of the revised manuscript (page 26, line 11-14).

7. I think that at the end of page 28, the Authors should refer also to the paper by Ye et al. (Cell 2016) as an example of full 3D activation mapping.

Comment #7: The reviewer recommended referring a paper by Ye et al. (Cell 2016) as an example of full 3D activation mapping.

We thank the reviewer for kindly introducing the critical reference. We refer to the study as a successful example of brain-scale neuronal activity mapping in the revised manuscript (page 29, line 16).

8 Did the Authors test any other antibody on human or marmoset? Is there any specific reason in the choice of the antibodies used (rather than e.g. NeuN)?

Comment #8: The reviewer asked the variation of tested antibody on human or marmoset and requested us to clarify the specific reason in the choice of the antibodies used on the primate samples.

We thank the reviewer for asking the point. The criteria for antibody selection in Fig.7 and Supplementary Fig. 7 were 1) compatibility with the three different species, 2) expectation of preserved or altered patterns of labeled structures for comparison, 3) constraints of IgG hosts and subtypes on the combination of multiple labeling, 4) sufficient signal intensity and penetration efficiency based on the results with mouse brains. In the current study, we only tested the combination of anti-GFAP and α-SMA antibodies

to meet the criteria. We addressed the point in our revised manuscript (page 80, line 16 – page 81, line 3). We will test a wider variety of antibodies on human tissues in our follow-up studies.

9. The gel/solution drawings in Fig. 1G are not very clear to interpret. I suggest the Authors to better distinguish free ions in the solution from ions attached to the gel – maybe using different colors.

Comment #9: Regarding Fig. 1g and Supplementary Fig. 1g, the reviewer suggested that free ions in the solution and gel-attaching ions should be more clearly distinguished by using different colors.

We appreciate the reviewer's kind suggestion. We changed the colors of free ions and gel-attaching ions in the revised Fig. 1 and Supplementary Fig. 1.

10. Scale bars in figures 2b-e or 3a-e. I guess that the actual size of gel or tissue will change according to different experimental conditions. Thus, each image would require its own scale bar.

Comment #10: The reviewer suggested that each image in Figs. 2b-e or 3a-e would require its own scale bar because the actual size of gel or tissue will change according to different experimental conditions.

We thank the reviewer's advice. We added a scale bar in all images in the revised Figs. 2b-e, 3a-e, Supplementary Figs. 2c and 3a. As expected (e.g., in Fig. 2b and c), the gel size was smaller at 500 mM than the 5 mM NaCl condition (thus the scale bar was larger). The result was consistent with Fig. 1g but with fewer dynamics, possibly due to the modulated osmolarity by HEPES or CUBIC-1A components.

11. Typo in fig. 3e: "appropreate" should be "appropriate"

Comment #11: The reviewer pointed out a typo in the panel of Fig. 3f.

We thank for the reviewer's comment. We corrected the typo in the revised Fig. 3f.

12. Methods, page 63, line 11-14. The patient from which the cerebellum was extracted by surgery is the same that suffered from amyotrophic lateral sclerosis? This is not clear in the current text.

Comment #12: The reviewer raised a question regarding the origin of the human cerebellum specimen used in Supplementary Fig. 7.

We thank the reviewer for pointing this out. In Supplementary Fig. 7, we used a fixed human cerebellum derived from an amyotrophic lateral sclerosis patient who underwent postmortem pathological dissection, not extracted by surgery. To more clarify the point, we rewrote the sentences in the revised manuscript (page 64, line 12-15).

Responses to Reviewer #2:

In “Versatile whole-organ/body staining and imaging based on electrolyte-gel properties of biological tissues”, Susaki et al. present work on developing an improved pipeline for staining and immunostaining cleared tissues using variations on the CUBIC methodology. The manuscript presents several principal findings: first, they show some relatively fundamental results that tissues (particularly brain) behave as a hydrogel using some classical physical chemistry assays (i.e., SAXS); second, they develop an assay for optimization of stain uniformity with small molecule stains of nucleic acids and antibody stains, and they use this assay together with their insights into tissues acting as a hydrogel to optimize conditions that favor uniform staining; third they demonstrate optimized labeling of a range of specimens. The high quality of the 3D images and demonstrations give an impression of a robust and useful method for the community. They are generally clear and convincing.

Overall, I think the work is well-suited for Nature Communications and would support its publication subject to the suggestions, below.

We are also glad that this reviewer was positively supported the publication of our study, mentioning that this manuscript demonstrates “the high quality of the 3D images” and “give an impression of a robust and useful method for the community”. We thank this reviewer for the very positive comments and successfully answered all the concerns in full.

1. In their electrolyte gel model, the positively charged nuclei dye, PI has strong interactions with the negatively charged gel and a high salt concentration improve the uniformity of labeling. The authors also demonstrate that high salt improves the uniformity of lipophilic dyes. Could the authors comment on the staining conditions likely to favor uniform labeling with a negatively charged dye? This would be interesting considering the authors’ statement that “most of the cationic amino residues had been masked by PFA fixation and the isoelectric points of CUBIC-delipidated brain was around pH 5, more abundant ionized residues in the polymer were thought to be the anionic carboxyl group.”

Comment #1: The reviewer asked a question regarding the staining condition which is likely to favor uniform labeling with a negatively charged dye, based on our statement that “most of the cationic amino residues had

been masked by PFA fixation and the isoelectric points of CUBIC-delipidated brain was around pH 5, more abundant ionized residues in the polymer were thought to be the anionic carboxyl group.”

We appreciate the reviewer for raising this important point. As stated in the manuscript, the positively charged dyes bound strongly to the surface of the negatively charged gel due to cationic-anionic interactions, which can be compromised by a high-salt condition. We hypothesized that this would not be the case for negatively charged dyes, because such interactions do not exist between the dyes and the gel. To test this assumption, we performed an additional gelatin gel assay and 3D staining of a delipidated cerebral hemisphere by using a representative negatively charged dye eosin (new Supplementary Fig. 2c, e). Indeed, the penetration of eosin into the gelatin gel or the tissue sample was independent of salt and ScaleCUBIC-1A chemicals. The 3D staining time was much shorter (~4 h for the cerebral hemisphere) than the case of nuclear stains (2-3 days at similar size). We additionally addressed the results and discussion in the revised manuscript (page 18, line 1-5; page 19, line 2-4; page 37, line 10-12).

2. I appreciate the authors' comments in the discussion about some limitations of this approach for the study of brains. However, I wonder whether other organs do not work as well as brains. For instance, in Fig. 2h, the nuclei stain from other organs is not as uniform as in brain. Could the authors comment on this?

Comment #2: The reviewer raised a concern that other organs would not work as well as brains. The reviewer pointed out an example in Fig. 2h, where the nuclei stain from other organs seemed not as uniform as in the brain.

We also thank the reviewer for pointing out the problems with the data displayed in the panel. We were concerned that the quality of the x-z reconstruction data in Fig. 2h might not be satisfactory for addressing the uniformity of staining due to the relatively large z-stack step size (100 μm). To more clarify the point, we provided a new movie (revised Supplementary Movie 1) of the LSFM x-y image. The data shows that all abdominal organs were stained and detected by LSFM imaging (note that the signals in the spinal canal are blurred because we had not cleared the bones by decalcification). Besides, related to the comment #1 of reviewer #1, we also added new experimental results of the swelling-shrinkage assay of the kidney, liver, and muscle (new Supplementary Fig. 1h) (page 14, line 9-17), as well as the whole-kidney staining with SYTOX-G and anti-CREB (new Supplementary Fig. 4p) (page 26, line 14-17). These results support the assumption

that other tissues resemble the brain in their physicochemical properties and 3D staining efficiency. We also discussed the extension of the insights found in the brain to other tissues (page 36, line 10-15; page 42, line 6-8).

3. Do the authors have any issues with substantial absorption of light through thick tissues? This is sometimes referred to as the “inner filter effect” and could potentially be quite strong for thick tissues.

Comment #3: The reviewer asked whether we have any issues with substantial absorption of light through thick tissues, so-called the “inner filter effect.”

We thank the reviewer’s question. At least in the whole mouse brain and the marmoset hemisphere, we did not observe such a practically problematic inner filter effect (please see Figs. 3j and 7c). We add new quantitation data to address the point (new Supplementary Fig. 3l) (page 24, line 7-8; page 92, line 1-7). On the other hand, we observed noticeable shadows at the medial side of the bones in the whole marmoset body image (Fig. 2g, h). We agree to the reviewer’s viewpoint regarding the issue and added some comments in Discussion of the revised manuscript (page 43, line 9-11).

4. Aspects of the presentation could be improved.

a. I feel the article was a bit longer than necessary to communicate the central points, but provide this feedback merely as a general suggestion.

Comment #4-a: The reviewer suggested that the article was a bit longer than necessary to communicate the central points and thus could be shortened.

We appreciate the reviewer's kind suggestion. This study covers several interdisciplinary topics to construct the whole story. Because biologists are not familiar with some of these topics, we have described as much detail as possible. To reflect the reviewer’s suggestion, we thoroughly removed redundant descriptions and rewrote some sentences with shorter phrases. However, in the revised manuscript, we minimally edited the first manuscript without significantly changing the text structure for the easiness of the revision process. We will finally shorten the main text more before publishing according to the journal guidelines and editor’s suggestions.

b. On P13L8 (page 13, line 8) the authors mention four-fold shrinkage of tissue, and they mean tissue volume, right? This should be made clear that it is not linear four-fold shrinkage which, on three axes, would be 64-fold shrinkage of tissue.

Comment #4-b: The reviewer suggested that we should make clear the degree of tissue shrinkage (Fig. 1g) more precisely.

We thank the reviewer for pointing this out. As described in Methods (Swelling-shrinkage assay of gels), We acquired bright-field images of the gels and the brain hemispheres, and then calculate their area values. The Y-axis of Fig 1g (S/S0) thus means the relative value of the gel area. A four-fold change of the S/S0 corresponds to an eight-fold change in volume. To more clarify the point, we added the relevant descriptions in the revised manuscript (page 13, line 7-9; legends in Fig. 1g and Supplementary Fig. 1g)

c. While this reviewer is sensitive that authors may not be native English speakers, there are nonetheless many typos or odd phrases. The language could use a fair amount of polishing (“Biological researches” on P6L5, “Does not penetration well” on P7L14, “Observed in the HEPES with” on P17L5, “Custom build” instead of custom-built on P18L3, and many others... I did not create a full list).

Comment #4-c: The reviewer suggested for us to correct typos or odd phrases in the manuscript.

We also appreciate the reviewer’s kind suggestion. We thoroughly checked and corrected our manuscript in the revised manuscript.

d. The results of this paper seem somewhat to contradict the results of the SWITCH approach from Kwanghun Chung’s group at MIT. The authors here state they couldn’t make SWITCH work. Perhaps there are aspects of the SWITCH protocol that haven’t been followed or that may not have originally been thoroughly described, but we are not in a position to judge from the brief statement. Overall the treatment of SWITCH feels a bit dismissive and critical without concretely explaining why. I think that a more careful discussion on this topic would be helpful and of great interest to practitioners. For instance, the following statement (P38L2) was vague in referring to SWITCH. “However, there were still potential concerns regarding on the two-step implementation and the choice of SDS without evaluation for broader antibodies in the study.” What were the

potential concerns to which the authors are now referring? In the SWITCH paper, they studied ~100 antibodies against ~40 different proteins, almost all of which were reported to work well with the SWITCH procedure. That is an equal or larger number of antibodies tested than in the manuscript under review, isn't it?

Comment #4-d: The reviewer suggested that a more careful discussion on the comparison of SWITCH would be helpful and of great interest to practitioners. The reviewer also commented that the authors of the SWITCH paper studied ~100 antibodies against ~40 different proteins, almost all of which were reported to work well with the SWITCH procedure.

We thank the reviewer for this comment. As in response to comment #5 of reviewer #1, we added our data for anti-NeuN 3D staining with SWITCH-mediated antibody labeling and CUBIC-HV in the new Supplementary Fig. 3j and k. Briefly, we performed the anti-NeuN 3D immunostaining corresponding to Figure 6C in Murray et al. Cell (2015) and compared it with CUBIC-HV. As shown in the new figures, we did not observe NeuN signals in the sample stained with SWITCH-mediated antibody labeling. In the revised manuscript, we described the results (page 23, line 17 – page 24, line 1) and discussed the potential issues of the tested SWITCH strategy more precisely, such as the insufficient recovery of immunolabeling in SWITCH-On buffer (PBST) (page 39, line 1-8). Apparently, the authors of the SWITCH paper were also aware of the issue and recently reported a new strategy for gradual environmental changes in their staining system [Yun et al. bioRxiv (2019), DOI: 10.1101/660373]. Regarding the antibodies used in the SWITCH paper, they used only the anti-Histone H3 antibody to demonstrate the implementation of SWITCH-mediated immunolabeling [in Figure 6C of Murray et al. Cell (2015)].

e. The quality of the movies was low. The authors should upload higher resolution movies that can properly show off their high-quality data. Hopefully there are no journal restrictions (such as low file size limits or poor default suggestions in the file preparation instructions) that are causing this.

Comment #4-e: The reviewer suggested that supplementary movies should be exchanged for higher resolution ones.

We thank the reviewer's kind suggestion. We replace them with their higher resolution versions (1080p).

Responses to Reviewer #3:

The present work presents a highly powerful and convincing 3D staining and imaging pipeline based on their previously established CUBIC approach. The results of this workflow, which the authors denote as CUBIC-HV is illustrated with many compelling experiments images.

The concept of describing the physicochemical properties of biological tissue as an electrolyte gel of cross-linked polypeptides is innovative, and the experiments leading to this conclusion are convincing. This is a particular strength of the current work that it rationalizes the otherwise often alchemistic art of 3D labeling. A clear advantage of the approach is that it can really quantify the soft material properties of tissue and then uses this knowledge to optimize labeling.

While I cannot judge all aspects of the work, in particular since I lack expertise in histo-chemistry and do not know exactly all lines of current state-of-the art, I am impressed by the breadth and depth of experiments and data collection, all aligned towards a well thought and research design of high intellectual merit. I can strongly recommend publication from a more general point of view of 3D histology. The approach may provide benchmarking results.

We are glad that this reviewer's overall enthusiastic comments, suggesting that CUBIC-HV is "highly powerful and convincing" and "innovative." We also thank the reviewer's understanding of our intention that by giving the concept of biological tissue as an electrolyte gel, this study has "a particular strength that it rationalizes the otherwise often alchemistic art of 3D labeling." We believe that we could fully answer all constructing questions and concerns raised by the reviewer.

At the same time, I would encourage the authors to avoid over-statements, and carefully check all statements made, mainly in view of (a) structure preserving, (b) necessary resolution for sub-cellular detail and cellular identification, imaging of brain circuitry.

At several instances when reading the work, I had strong impressions of over-claims. For example, concerning cellular identification.

1) The voxel sizes are quite large in many of the results, and resolution was not determined at all (for example by FSC). This is clearly a deficit of the MS. The rather modes resolution must affect the recall and precision of cellular identification. Please comment this critically, wherever applicable in the MS.

Comment #1: The reviewer raised a concern that the voxel size and resolution of the imaging results are quite large and affect the recall and precision of cellular identification. The reviewer also recommended for us to address the point by determining the image resolution in the manuscript.

We thank the reviewer for commenting on the point. The typical voxel size of the data acquired by GEMINI microscopy (Fig. 3-6) was 8.3 or 6.5 μm for x-y and 9 or 7 μm for z (used 0.63 \times objective with 1.25 \times or 1.6 \times optical zoom, respectively) as described in the figure legends. Here, the voxel size was calculated by the pixel size of the sCMOS sensor (6.5 $\mu\text{m}^2/\text{pixel}$) and the step size of z-axis (9 or 7 μm). Given the theory of Nyquist frequency, the resolving power based on the voxel size can be determined as 16.6 or 13 μm for x-y and 18 or 14 μm for z. Since the value was larger than the optical resolution of the microscopy (x-y: approximately 6.6 or 4.2 μm at $\lambda = 550 \text{ nm}$, provided by Olympus, z: approximately 10 μm at $\lambda = 550 \text{ nm}$, based on the effective N.A. for light-sheet illumination ~ 0.03), the voxel size of the images can account for the resolution of the data.

Because CUBIC-R+ makes the sample swell about 1.5-folds, the voxel resolution is corresponded with about 11.1 or 8.7 μm for x-y and 12 or 9.3 μm for z before clearing. The resolution range is not “quite large” but similar to the cell body and nucleus size. It was thus enough for resolving each nucleus in the data (Fig.3j) as well as computationally detecting sparsely labeled cell body/nuclei (Supplementary Fig. 6). We additionally describe these points more precisely in the revised manuscript (page 89, line 15 – page 90, line 10).

As commented in response to comment #4 of reviewer #1, we think that the computational cell detection results here are sufficient for showing the immunolabeling and imaging quality by CUBIC-HV. We also recognize the users’ demand for a more sophisticated whole-organ cell detection and analysis method for which we will address in our future manuscript.

2) Closely related to this point: p. 29, l.13 and following: why does this experimental result prove identification of whole brain circuit structure, neuronal connections cannot be visualized in this manner.

Comment #2: The reviewer pointed out a possible inappropriate description at p. 29, line 13 of the original manuscript, asking why the experimental results in Fig. 6a-j prove the identification of the whole-brain circuit structure.

We also thank the reviewer's comment. The rabies virus tracing system has been widely used in the neuroscience field to label the whole-brain neuronal circuits, which relies on the retrograde transsynaptic activity of the viral species. This system has been well documented and used in many studies (e.g., Osakada and Callaway, Nature Protocol 2013). Thus, our data capturing the whole brain image of the RV-labeled sample (Fig. 6b-c) prove successful visualization of whole-brain neuronal circuits in the 3D data. We focused on the injection site for the detailed analysis of multi-labeled cell identification because the GABAergic connections (including Sst-positive neurons) out of other cortical regions (e.g., somatosensory cortex) are rare in the Gad2-Cre strain [Fig. 6e; mouse connectivity map of Allen brain atlas, #167441329 (AAV-labeled projection S1 barrel of Gad2-IRES-Cre strain), <http://connectivity.brain-map.org>, also see the reply to comment #4 of reviewer #1] (page 97, line 16 – page 98, line 10). To respond to the reviewer's comment, we modified the sentence with more specified content in the revised manuscript: "These results indicate that the neural circuitry and cell types could be visualized with single-cell resolution in the whole-brain multimodal labeling data" (page 30, line 10-12).

3-1) p 42, l.17.18, Suppl. Fig.4: Please explain more explicitly to which end the data shown in Fig.4 proves preservation of subcellular structure and molecular integrity.

Comment #3-1): The reviewer asked that which data appeared in Fig.4 (p 42, line17-18 in the original manuscript) prove preservation of subcellular structure and molecular integrity.

We thank the reviewer for asking the point. The description of p42 corresponds to the contents of p25, line 12 - p26, line 2 of the original manuscript. We have shown the preservation of subcellular structure and molecular integrity (i.e., subcellular localization of the molecules) in Supplementary Fig. 4j-o. If they had not been preserved, these structures (e.g., axons, dendrite and spines) would not be observable in the samples. To more clarify the point, we added the reference to these figure panels in the revised manuscript (page 43, line 8).

3-2) Another example, where I am skeptical to which end neurons are well identified is the example of human cerebellum.

Comment #3-2): The reviewer raised a question regarding a neural cell type stained in the human cerebellum (Supplementary Fig. 7).

We again thank the reviewer's question. We stained the human cerebellar sample with GFAP (glial fibrillary acidic protein, a representative marker for astrocytes) and α -SMA (alpha-smooth muscle actin, a representative marker for arterial smooth muscle) antibodies, and we did not visualize any neural cells in the sample. These two antibodies are widely used for detecting these cell types. The staining results were considered reasonable by the experts among the authors.

4) Suppl- Fig.7. Please explain at which level (recall, precision, dice score etc.) neurons in the molecular, Purkinje cell layer, and granular layer can be identified.

Comment #4: The reviewer also raised a question that at which level (recall, precision, dice score etc.) neurons in the molecular, Purkinje cell layer and granular layer can be identified in the data of Supplementary Fig. 7.

We also appreciate the reviewer's question related to the comment #3-2). As answered above, the data does not include any information about neuronal cell types that allows to quantitatively deal with the level of neuronal identification in each cerebellar layer. The cerebellar structure has been well studied and appears in many textbooks such as Carpenter's Human Neuroanatomy by Parent and Carpenter (published by Williams & Wilkins). Therefore, the cerebellar layers can be easily identified on histological images. To address the reviewer's request and as a reference for readers, we add a new Klüver-Barrera (K-B) staining image acquired from the same human specimen (new Supplementary Fig. 7e). Related to the response to comment #2 of the reviewer #1, we also updated the result paragraph of the figure (page 33, line 11 - page 34, line 9).

5) SAXS curves of non-delipidated and delipidated tissue. The tentative attribution of the peak at 0.08 rec. Angstrom to myelin should be substantiated by comparison to literature, for example Carboni et al, Biomedical Optical Express 2017, who report a first peak at about 0.09 and a second at 0.16. There may be interesting

conclusions/explanations regarding the osmotic pressure and other variable which may affect the d-spacing of myelin here.

Comment #5: The reviewer pointed out that the tentative attribution of the peak at 0.08 μm to myelin should be substantiated by comparison to literature, for example, Carboni et al, Biomedical Optical Express 2017, who reported the first peak at about 0.09 and the second at 0.16.

We appreciate the reviewer for suggesting this excellent paper, which addresses the peak as the signal of myelin sheath in the SAXS measurement of the brain. We hypothesize that the differences in species (human vs. mouse) and sample preparation methods (dried sample vs. PFA-fixed sample in buffer) in the two studies may cause a slight difference in the measured values (0.09/0.08). In the revised manuscript, we focused on the similarity of peak positions in the two studies and suggested that the scattering peaks found in non-delipidated tissues could be attributed to the myelin sheath, citing the recommended paper (page 11, line 10-18).

6) The interpretation of the SAXS curves in terms of a fractal geometry should be rephrased more carefully, which a fractal structure can explain the power laws, it cannot be uniquely be inferred from the data. A model and full q -range fit of the data would be required to really exploit the information in this data.

Comment #6: The reviewer also pointed out that the interpretation of the SAXS curves in terms of a fractal geometry should be rephrased more carefully.

We thank the reviewer's comment. Since the sample was PFA-fixed and composed of cross-linked polypeptides, we postulated that the tissue sample should surely be considered as a monolithic gel material in which various sizes of the mesh can coexist. As a result, the power-law in the I - q plot was observed, which meant the fractal scattering pattern reflecting a diverse range of mesh sizes in the sample (Schaefer, Science 1989; Beaucage, J Applied Crystallography 1996). While we think that the collection of data in Fig. 1 supports our claim that the tissue sample is a gel with a fractal structure, we agree to the reviewer's argument that the I - q plot data should be more carefully interpreted in a part of our logic. To respond to the reviewer's comment, we corrected and rephrased some of the related sentences in the revised manuscript (page 12, line 11-17; page 36, line 2-9; legend of Fig. 1e).

7) The diffusion-reaction model of the staining process: Please add information of time scale and time units (simulation steps). A homogeneous distribution is clearly the steady-state solution of the equation, so all non-homogeneous pattern should only be solution while the systems converges to a steady state. Time scales (even if only in 'simulation units') is critical here and need to be described.

Comment #7: The reviewer requested us to add information of time scale and time units (simulation steps) in the diffusion-reaction model of the staining process because all the non-homogeneous patterns should only be observed in the solution while the system converges to a steady state.

We agree with the reviewer that the simulation condition is critical for the evaluation of our simulation results. We additionally described the detail of the simulation process in Method section (page 75, line 1-7) and the legend of Supplementary Fig. S2a of the revised manuscript.

8) While it is very well written, typos and grammar have to be checked throughout the MS.

Comment #8: The reviewer recommended us to check typos and grammar throughout the MS.

We appreciate the reviewer's kind advice. We thoroughly checked and corrected the sentences in the revised manuscript.

REVIEWERS' COMMENTS:

Reviewer #1 (Remarks to the Author):

Dear Editor,

all my concerns have been satisfactorily addressed in the current version of the paper. I thus suggest its acceptance for publication.

My only suggestion on the revised manuscript is to include the data about sensitivity and PPV of the cell detection in the Methods section. This information is currently only in the Legend of a Supplementary Figure, not very easy to find.

Reviewer #2 (Remarks to the Author):

Their revision does a good job addressing my original feedback and I now support its publication.

Some odd phrases and grammar errors persist, but these should be easily corrected during the finalization of the manuscript.

Reviewer #3 (Remarks to the Author):

The authors have responded convincingly to all questions and issues raised. I can recommend publication.

Responses to Reviewers' Concerns for Susaki et al., "Versatile whole-organ/body staining and imaging based on electrolyte-gel properties of biological tissues".

We were delighted that all three reviewers supported the publication of our manuscript. We thank for their comments, especially for the reviewer #1's vital suggestion to include the data about sensitivity and PPV of the cell detection in the Methods section, and for the reviewer #2's recommendation to further correction of grammar and phrases. We are confident that we have addressed all the reviewers' concerns in the revised manuscript, of which details are described as follows.

Reviewer #1 (Remarks to the Author):

All my concerns have been satisfactorily addressed in the current version of the paper. I thus suggest its acceptance for publication.

My only suggestion on the revised manuscript is to include the data about sensitivity and PPV of the cell detection in the Methods section. This information is currently only in the Legend of a Supplementary Figure, not very easy to find.

We thank the reviewer for their support for publication. As suggested by the reviewer, we additionally described the data of sensitivity and PPV of the cell detection algorithms (in "Computational cell detection" of the Methods section).

Reviewer #2 (Remarks to the Author):

Their revision does a good job addressing my original feedback and I now support its publication.

Some odd phrases and grammar errors persist, but these should be easily corrected during the finalization of the manuscript.

We also thank the reviewer for their support for publication and suggestions for further correction. Our manuscript has been finally checked and edited by Nature Research Editing Service, according to the editor's recommendation.

Reviewer #3 (Remarks to the Author):

The authors have responded convincingly to all questions and issues raised. I can recommend publication.

We appreciate for the reviewer's recommendation for publication.